# Sensing and memorising liquids with polarity-interactive ferroelectric sound

Jong Sung Kim[1,3], Eui Hyuk Kim[1,3], Chanho Park[1], Gwangmook Kim[1], Beomjin Jeong[1], Kang Lib Kim[1], Seung Won Lee[1], Ihn Hwang[1], Hyowon Han[1], Seokyeong Lee[1], Wooyoung Shim [1], June Huh[2] & Cheolmin Park[1]

The direct sensing and storing of the information of liquids with different polarities are of significant interest, in particular, through means related to human senses for emerging bio-medical applications. Here, we present an interactive platform capable of sensing and storing the information of liquids. Our platform utilises sound arising from liquid-interactive ferro-electric actuation, which is dependent upon the polarity of the liquid. Liquid-interactive sound is developed when a liquid is placed on a ferroelectric polymer layer across two in-plane electrodes under an alternating current field. As the sound is correlated with non-volatile remnant polarisation of the ferroelectric layer, the information is stored and retrieved after the liquid is removed, resulting in a sensing memory of the liquid. Our pad-type allows for identifying the position of a liquid. Flexible tube-type devices offer a route for in situ analysis of flowing liquids including a human serum liquid in terms of sound.

[1] Department of Materials Science and Engineering, Yonsei University, Seoul 03722, Republic of Korea. [2] Department of Chemical and Biological Engineering, Korea University, Seoul 02841, Republic of Korea. [3]These authors contributed equally: Jong Sung Kim, Eui Hyuk Kim. Correspondence and requests for materials should be addressed to C.P. (email: cmpark@yonsei.ac.kr)

The development of electronic interfaces for humans to quantitatively gain and store the information of things including hazardous gases, chemical and biological liquids and powders that are readily exposable to humans is of great importance for the emerging biomedical and health care applications based on Internet of Things technologies[1–8]. The direct sensing, monitoring and storing of the information of liquids[9–12] having different polarities are significant challenges, in particular, through means related to human senses such as vision and smell[13–19]. In general, it is extremely difficult for human senses to distinguish two liquids that are mostly colourless and transparent, except through identification based on their unique smell, which is rarely recommendable owing to the possible toxicity of volatile liquids. Solvatochromic[20–23] materials are capable of changing their optical absorbance in the visible range depending upon the polarity of the liquid containing them, allowing human eyes to see the difference in liquid polarity. As these vision-related liquid detections further require bulky spectroscopic facilities for detailed analysis of the polarity-dependent absorbance, development of a wearable and/or on-body device platform useful for human–things interaction is hardly achievable with solvatochromic materials. Liquid sensing based on electro acoustical techniques is also promising, but the acoustic frequencies used the previous works are not audible from MHz to GHz, which region requires special detection facility[24–28]. More importantly, high volatility of most liquids in air limits their detection and analysis time. Therefore, it is essential to develop a liquid sensing platform in which the information of a liquid can be stored in a non-volatile manner and extracted many times even after removal of the liquid from the platform.

Here, we present an interactive sensing memory platform capable of sensing, monitoring and storing the information of various liquids. Our platform utilises sound arising from liquid-interactive ferroelectric actuation, which is dependent upon the polarity of the liquid, allowing for direct sensing and storing of information of various liquids. Liquid-interactive ferroelectric sound (LIFS) is successfully developed when a liquid droplet is placed on a ferroelectric polymer layer across two in-plane electrodes underneath the ferroelectric layer under an external in-plane AC field. An AC field built up vertically between one of the electrodes and the liquid, varying with the polarity of the liquid, results in film actuation arising from the AC-field-dependent ferroelectric polarisation of the film. The sound pressure level (SPL) of a device, in turn, depends upon the polarity of the liquid, allowing for facile liquid sensing and identification. More importantly, as the SPL arising from LIFS of a liquid is correlated with the non-volatile ferroelectric remnant polarisation of the vibrating layer, the information of a liquid is readily stored and retrieved even after the liquid is removed, resulting in a sensing memory of the liquid. By employing a microfluidic channel on our platform, the flow of a human serum liquid through the channel is monitored and its velocity is obtained in terms of SPL. We also demonstrate that LIFS is useful for identifying the 2-D position of a liquid droplet on a thin pad-type device with position-addressable LIFSs. Furthermore, mechanically flexible tube-type LIFS AC devices allow for in situ sensing of a fluid passing through the tube in terms of SPL.

## Results

**Fabrication of a LIFS**. Our LIFS AC device was fabricated through all-solution-based processes as shown in Fig. 1a (Supplementary Table 1 and Supplementary Fig. 1). The architecture of the LIFS AC device consists of three polymer layers. In brief, two in-plane electrodes of 400-nm-thick poly(3,4-ethylenedioxythiophene) polystyrene sulfonate (PEDOT:PSS) were spin-coated on a clean glass substrate and subsequently, reactive ion etching (RIE) was performed for separating the two electrodes. A buffer layer of poly(methyl methacrylate) (PMMA) and a ferroelectric layer of poly(vinylidene fluoride-co-trifluoroethylene) (PVDF-TrFE)[29–31] of thickness ~1.2 and 5.0 μm, respectively, were sequentially spin-coated on the in-plane PEDOT:PSS electrodes. The fabricated planar-type device was employed for liquid sensing and memory with liquid deposited on the device. A tube-type device was fabricated by sequential spin-coating of PVDF-TrFE and PMMA on a Si substrate, followed by the deposition of in-plane PEDOT:PSS electrodes. When detached from the substrate, the three layered platform was readily deformed and a tube-type device was developed as shown in Fig. 1a. The PMMA layer was employed to flatten the surface of the PVDF-TrFE layer.

**Principle of liquid sensing and memory based on LIFS**. First, we observed that half of the AC voltage[32–35] exerted between the two in-plane PEDOT:PSS electrodes was developed between one of the PEDOT:PSS electrodes and a metallic layer (e.g. Al and Au) deposited on the PVDF-TrFE layer of the LIFS AC device (Supplementary Fig. 2). The results were consistent with those obtained in our previous organic light-emitting board[32]. Subsequently, the vertical voltages between the bottom PEDOT:PSS electrode and various liquids deposited on the LIFS AC device were measured, as representatively shown in the photographs of Fig. 1b. When a conductive PEDOT:PSS solution was placed on the PVDF-TrFE surface, the vertical voltage of ~47 V was measured, which was slightly lower than half of the applied in-plane voltage (50 V) possibly owing to its lower conductance than that of either Al or Au as shown in Fig. 1c (Supplementary Fig. 2). Interestingly, the voltage built up vertically between one of the bottom PEDOT:PSS electrodes and a floating substance was significantly dependent upon the polarity of liquids and the voltage decreased with the decrease in the polarity of the liquids, as shown in Fig. 1c. Notably, the voltage that developed vertically between a PEDOT:PSS electrode and liquid approached zero at frequencies greater than several hundred kilohertz; this frequency field is too high for PVDF-TrFE to respond[36]. The voltage drop at certain frequencies was ascribed to a decrease in the dielectric constant of the PVDF-TrFE layer at the frequencies (Supplementary Fig. 2d). The vertical voltage induced with a polar liquid from an LIFS device was also carefully measured and averaged with our device characterisation system using 10 sets of LIFS devices (Supplementary Fig. 3).

To elucidate the electric field developed in our LIFS device, we first performed an electric field calculation based on the finite element method (FEM), with which we were able to confirm the vertical field arising from a top conductive layer; the results are shown in Fig. 1d, e. A typical in-plane and fringe field was developed between two in-plane electrodes on which PMMA and PVDF-TrFE layers were placed without a top conductive layer. When a top conductive layer was deposited, a vertically driven electric field developed on the two overlapped regions of the top conductive layer with the two in-plane electrodes. Notably, the vertical fields on the two overlapped areas were opposite in direction, which implies that the top conductive layer acted as a type of electric field mirror. Further experiments also revealed that the vertical field depended upon electron as well as ion conductivity (Supplementary Fig. 4).

In our LIFS device, two insulating layers of PVDF-TrFE and PMMA are connected in series between one of the two in-plane electrodes and a liquid. Based on the dielectric constants and the thickness of the PVDF-TrFE (5 μm) and PMMA (1.2 μm) layers, we were able to calculate the electric field exerted on the PMMA layer, which was ~30% of the applied electric field. The electric field distribution of a device with a PMMA/PVDF-TrFE bilayer

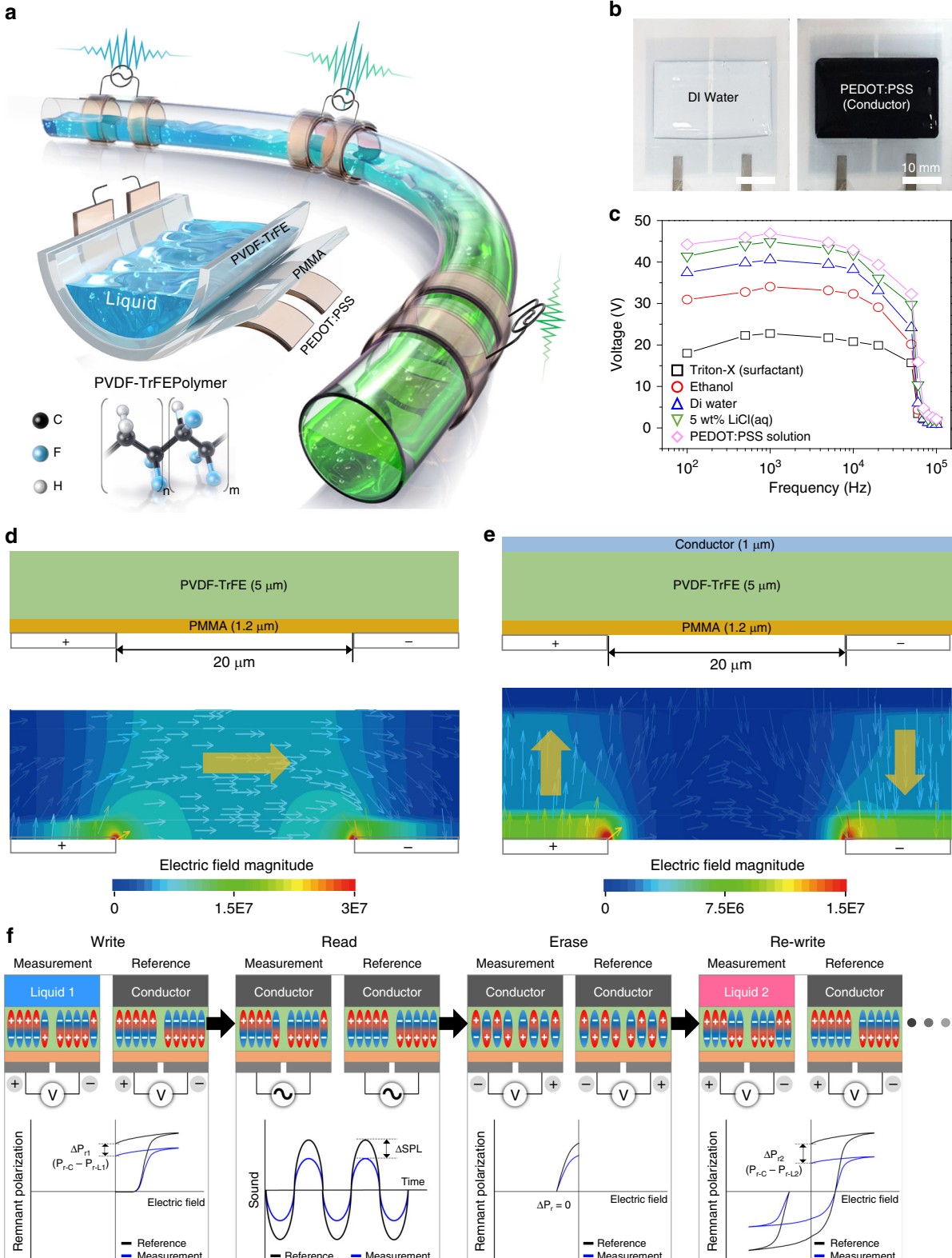

**Fig. 1** Device architecture and working principle. **a** Conceptual illustration of liquid-interactive ferroelectric sound (LIFS) in our tube-type alternating current (AC) device containing a ferroelectric PVDF-TrFE tube with two pairs of in-plane electrodes on the surface of the tube. LIFSs with different sound pressure levels (SPLs) are developed using liquids with different polarities. A schematic of a flexible planar-type LIFS AC device is also shown with three layers of two in-plane PEDOT:PSS electrodes. **b** Photographs of planar-type LIFS AC devices with deionised water and PEDOT:PSS solution in water placed on the devices. **c** AC voltages measured between one of the bottom PEDOT:PSS electrodes and various liquids deposited on an LIFS AC device as a function of frequency at the voltage of 100 V. Finite element method (FEM) results of the LIFS AC device under a voltage bias between two in-plane electrodes showing both the direction and magnitude of the generated electric field without (**d**) and with (**e**) a top conductive layer on PVDF-TrFE. **f** Schematic illustration of non-volatile writing, reading, and erasing of the information of a liquid with an LIFS AC device in terms of liquid-polarity-dependent ferroelectric polarisation, which can be in turn converted into non-volatile SPL

clearly shows a vertical electric field in the PVDF-TrFE layer, although the field strength is somewhat reduced owing to the presence of the PMMA layer (Supplementary Fig. 5). By considering the vertical electric field shared by the PVDF-TrFE and PMMA layers, we were able to propose a simple linear electric field model to describe the polarity-dependent vertical electric field exerted on the PMMA and PVDF-TrFE layers (Supplementary Fig. 6).

The liquid-polarity-dependent voltage built up on our LIFS AC device offered a facile route for identifying the liquids by means of the sound arising from the ferroelectric vibration of the PVDF-TrFE layer, which was also dependent upon the vertical AC voltage as shown later. Furthermore, as the remnant polarisation ($P_r$)[37–41] of the ferroelectric layer was dependent upon the applied voltage, the LIFS AC device enabled us to retrieve the information of a liquid stored in terms of liquid-dependent $P_r$ even after removing the liquid as schematically shown in Fig. 1f. First, $P_r$ was programmed with Liquid 1 having a certain polarity placed on the PVDF-TrFE layer whereas a reference device was poled with a floating conductive layer instead of the liquid (the write step of Fig. 1f). One could readily assume that the $P_{r-C}$ with the conductor was greater than that with Liquid 1 ($P_{r-L1}$). The information of Liquid 1 programmed with $P_{r-L1}$ was read under an AC field in terms of sound after removing the liquid. To retrieve the SPL arising from $P_{r-L1}$, a reference conductive layer should be placed on the programmed PVDF-TrFE layer. The difference in SPL between the device arising from $P_{r-L1}$ and that from $P_{r-C}$ was measured in time domain (the read step of Fig. 1f). The programmed $P_r$ from either the liquid or the conductor was erased by applying a reverse voltage sufficiently larger than the coercive field of the PVDF-TrFE layer, making the dipoles in both devices randomised as shown in the erase step of Fig. 1f. After removing the conductive layer in the measurement device, another liquid, Liquid 2, was placed, followed by the application of the programme DC voltage. This re-write process developed $P_{r-L2}$ dependent upon the polarity of Liquid 2. The difference in SPL was characterised in the read process again, yielding a non-volatile, re-writable liquid sensing device.

**Optimisation of device architecture and sound performance**. The device architecture and dimensions of the components including the spacing between the two parallel electrodes and the relative area of the two electrodes of the planar-type LIFS AC device, schematically shown in Fig. 2a, were optimised (Supplementary Figs. 7 and 8). Our home-built SPL detection system was employed where SPL values arising from LIFS of a liquid were obtained as a function of applied AC frequency and voltage, as shown in Fig. 2a. The sensing performance of a liquid on the LIFS AC device based on SPL was enhanced by maximising the difference in SPL before and after liquid deposition. In our device architecture, we obtained the greatest SPL ON/OFF difference when a PMMA layer of thickness ~1.2 μm was inserted between the PVDF-TrFE layer and a PEDOT:PSS electrode (Supplementary Fig. 7a). To enhance the LIFS, which strongly depends upon the piezoelectric polarisation of PVDF-TrFE, the electric poling process was employed (Supplementary Fig. 7d). The thickness of the PVDF-TrFE film was optimised to ~5 μm under the applied DC poling voltage of 2 kV. One of the advantages of a thin PVDF-TrFE film is that the LIFS AC device is uniformly generated regardless of the detection position. This detection-angle-independent sound generation was confirmed with our LIFS AC device on a rotating stage (Supplementary Fig. 7e).

**Polarity-dependent LIFS**. For a systematic study of the dependence of the vertical AC voltage on the polarity of liquids, we examined two sets of dielectric and ionic liquids with a planar-type AC device, and the results are shown in Fig. 2b–e. We extensively examined 18 solvents having different polarities; the polarity-dependent SPL results are shown in Fig. 2c (Supplementary Table 2). The SPL is almost linearly proportional to the dielectric constant values of the solvents, and the sensitivity of an LIFS device, which was obtained from the slope of the curve, is ~0.75 (dB/$\varepsilon_r$). The SPL values also increased with the ionic conductivities of the liquids, which were controlled by the concentration of lithium chloride (LiCl) in deionised water; however, the conductivity dependence was small, as shown in Fig. 2d (Supplementary Fig. 4c). The ON/OFF ratios of the SPLs of the liquids with different polarities in Supplementary Fig. 9 indicate that our LIFS AC device is suitable for sensing various liquids. It should be noted that the sensing of the good solvents such as THF, MEK, pyridine, DMF and DMSO should be performed by employing a protective layer insoluble in the solvents (Supplementary Fig. 10).

The frequency-dependent SPLs of the liquids were examined as a function of AC frequency ranging from 20 Hz to 20 kHz and the results are shown in Fig. 2e. The SPL value increased with the frequency mainly owing to the enhancement of the ferroelectric vibration with the frequency. A greater frequency dependence of SPL was observed at higher polarity of liquids, making the identification of liquids easier at higher frequencies. The results showed that the liquids were identified in terms of SPL over the full frequency range typically audible to the human ear. The liquids were distinctly identified in terms of SPL over the broad range of the applied voltage, as shown in Supplementary Fig. 12a. The results indicate that the device can be successfully operated at a voltage as low as 20 V with distinct sound-based sensing capabilities (Supplementary Fig. 12a, Supplementary Movie 1 and 2).

In our LIFS device, the vertical field distinctly depends on the overlapped area. The SPL values increased with the contact areas (Supplementary Fig. 13). Interestingly, the electric field rarely changes when the heights of the liquid is varied in the range from a few micrometres to a few millimetres (Supplementary Fig. 14). To make our SPL results consistent, we set up a reservoir with a constant contact area as schematically shown in the Fig. 2a. Note that the heights-independent liquid sensing of our LIFS device makes our approach different from that of the conventional film bulk acoustic resonator (FBAR) associated with a piezoelectric resonance frequency sensitive to the mass loading placed on the layer[42–44].

The $d_{33}$ of a PVDF-TrFE layer is aligned vertically, and is therefore responsible for the ferroelectric polarisation which depends on the polarity of liquids (Fig. 2). The vertical alignment of the $d_{33}$ mode of a PVDF-TrFE film in our LIFS device was additionally confirmed with a piezoresponse force microscope (PFM) (Fig. 3). PFM images show that the piezoelectric amplitude significantly varies on the two overlapped regions, compared with the gap area. On the other hand, no piezoelectric response is observed on the PVDF-TrFE film when an in-plane field was applied without a top conductive layer. As expected from the FEM results, which revealed that the field directions of the two overlapped areas were opposite each other (Fig. 1f), the piezoelectric response in the two overlapped regions is opposite in amplitude, as shown in the amplitude profile shown in the schematic ferroelectric dipoles in the corresponding regions in Fig. 3c, d. The results shown in Fig. 3 clearly demonstrate that the higher the amplitude developed, the more polar a liquid is, which is consistent with the sound memory results in Fig. 4b.

The dipole orientations in various liquids on the ferroelectric polymer layer were also examined by the molecular dynamic (MD) simulations (Supplementary Fig. 15) from which we computed the time-dependent electric dipole moments for five

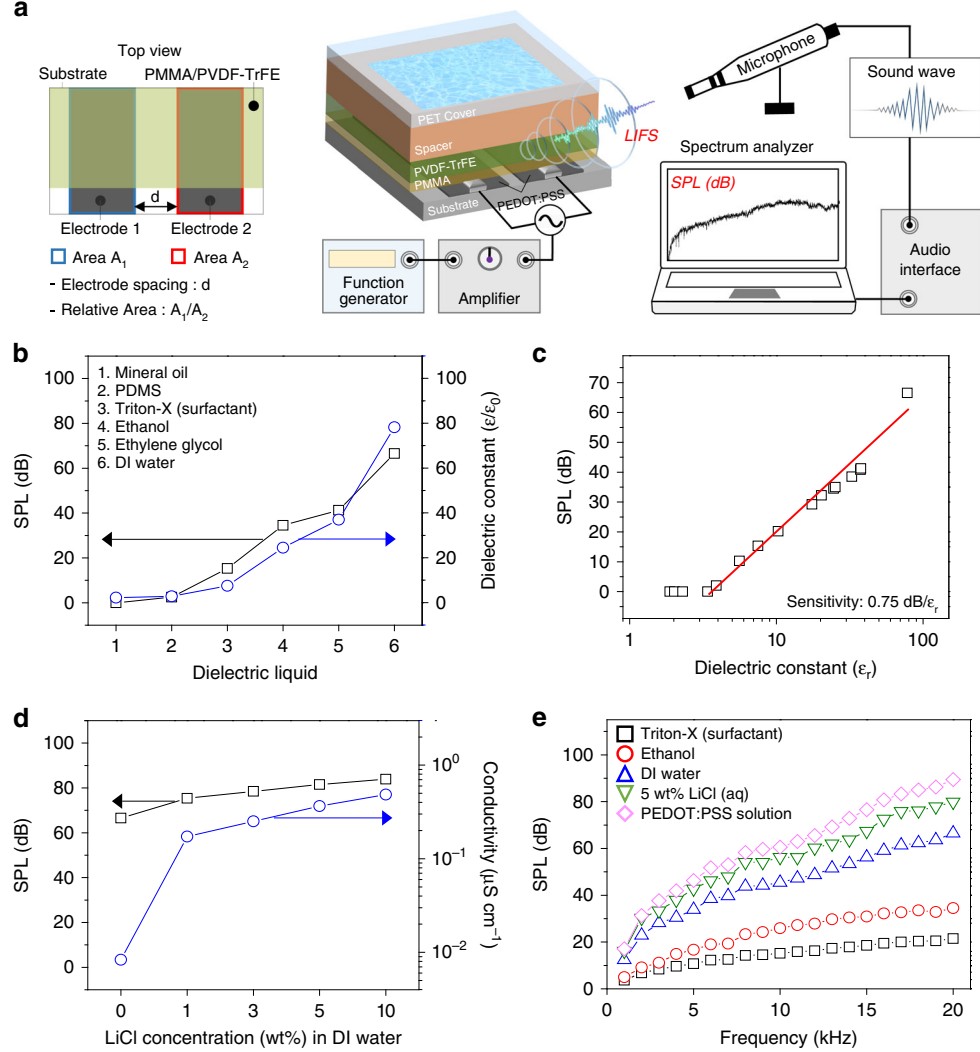

**Fig. 2** Sound performance from liquid polarity-interactive ferroelectric actuation. **a** Schematics of architecture and sound pressure level (SPL) detection system of the planar-type liquid-interactive ferroelectric sound (LIFS) alternating current (AC) device. SPL values arising from LIFS of a liquid were obtained as a function of applied AC input signal. **b** SPL values of six liquids examined with different dielectric constants on an LIFS AC device. The SPL values are proportional to the dielectric constants of the liquids. The device was operated at an AC frequency and voltage of 20 kHz and 100 V, respectively. **c** SPL values of 18 liquids examined with different dielectric constants, and the sensitivity of an LIFS AC device obtained from a linear slope of the curve. The device was operated at an AC frequency and voltage of 20 kHz and 100 V, respectively. **d** SPL values of five liquids examined with different ionic conductivities controlled by the amount of LiCl in water on an LIFS AC device. SPL values increased with the ionic conductivities of the liquids. The device was operated at an AC frequency and voltage of 20 kHz and 100 V, respectively. **e** SPL values of five liquids with different polarities on an LIFS AC device as a function of AC frequency at the AC voltage of 100 V

different liquids (water, acetonitrile, isopropanol, chlorobenzene and cyclohexane) placed on PVDF crystal substrate. The time-dependent dipole moments in the directions parallel to the PVDF substrate ($\mu_x$, $\mu_y$) fluctuate around zero and these dipole fluctuations in-plane direction are more pronounced as the liquid becomes polar, which indirectly represent the susceptibility of molecules in the liquid towards their dipole orientation by an external vertical field, consistent with the PFM results. The integration and scalability of the device are also important for further development. We systematically examined the device performance by controlling the size of the water droplets in an LIFS device with a fixed in-plane electrode gap of 0.2 mm. The results (Supplementary Fig. 16) show that our LIFS device successfully operates with a water droplet of minimum diameter ~1 mm. Further scaling-down of our device can be achieved by reducing the gap between the two in-plane electrodes (Supplementary Fig. 17).

**Non-volatile LIFS AC memory**. The polarity-dependent SPL of a liquid was also memorised in terms of the polarity-dependent $P_r$ of the PVDF-TrFE layer and the information was successfully retrieved even after removing the liquid. For the liquid sensing memory performance of our LIFS AC device, both DC and AC inputs were required for writing and reading processes, respectively, as shown in Fig. 4a. Notably, however, the writing and reading steps were both accomplished with a single circuit design. To guarantee constant volume of the liquid during the measurement, the LIFS AC device was modified with a fluidic channel[45–48] of 1-mm-high VHB spacers. As described earlier, a DC programming voltage of 2 kV, which is sufficiently higher than the saturation voltage of a PVDF-TrFE layer with a reference PEDOT:PSS floating electrode, was applied on the LIFS AC device with various liquids (Supplementary Fig. 18). Notably, our parallel-type LIFS AC device required voltage two times greater than that for polarisation saturation of vertical device as shown in

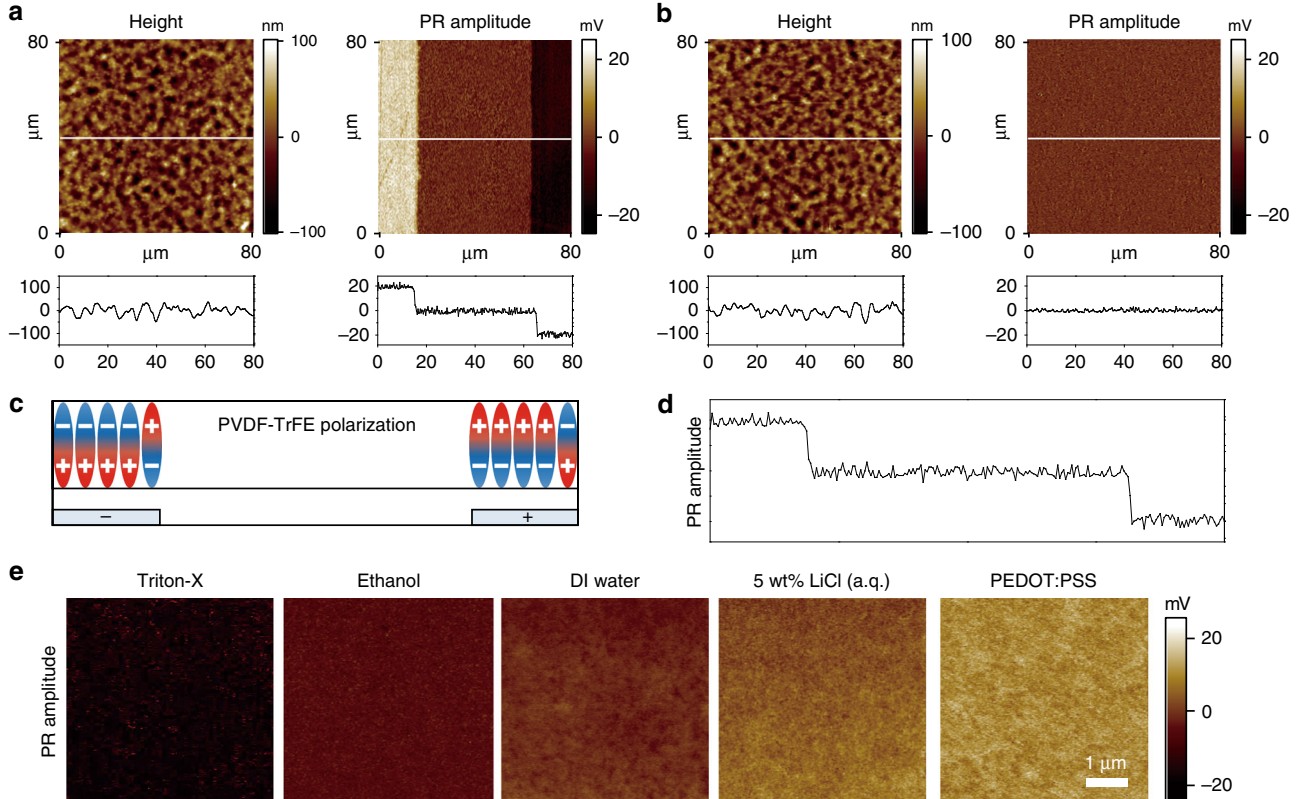

**Fig. 3** Vertically aligned ferroelectric polarisation of the PVDF-TrFE film. Piezoresponse Force Microscopy (PFM) images in height and piezoelectric amplitude contrast of a PVDF-TrFE layer in the liquid-interactive ferroelectric sound (LIFS) alternating current (AC) device after direct current (DC) poling between the two in-plane electrodes with (**a**) and without (**b**) a top conductive layer. Distinct piezoelectric amplitudes with opposite contrasts are apparent in the two overlapped regions between the top conductive layer and one of the two electrodes. **c** Schematic of the ferroelectric dipoles of a PVDF-TrFE layer developed in the two overlapped regions under DC poling between the two in-plane electrodes. **d** Amplitude profile along the horizontal line in **a** across the two overlapped regions. **e** PFM images in the piezoelectric amplitude contrast of PVDF-TrFE layers programmed with five liquids having different dielectric constants placed on the PVDF-TrFE film in an LIFS AC device

Supplementary Fig. 19. The five representative liquids with different polarities were programmed with the DC voltage and subsequently removed from the devices. The $P_r$ programmed in the PVDF-TrFE layer dependent upon the polarity of liquids was successfully converted into SPL when the devices were operated with the reference PEDOT:PSS electrode. The results shown in Fig. 4b (Supplementary Fig. 21a) indicate that SPL values of the liquids were proportional to the polarity of the liquids.

The SPL arising from the $P_r$ of a liquid was erased by depolarising the remnant dipoles in the PVDF-TrFE layer with the reverse electric field of −1 kV. For instance, the SPL of approximately 65 dB resulting from the $P_r$ of water was significantly reduced to ~38 dB after the removal of the $P_r$, as shown in Fig. 4c. The re-programming of the $P_r$ with water yielded a high SPL of ~65 dB when read by the reference PEDOT: PSS electrode. Our LIFS AC device exhibited reliable programme-read-erase cycles over at least 10 times as shown in Fig. 4c. In addition, we examined the write-read cycle endurance of an LIFS AC device with two different liquids upon multiple cycles of consecutive write with deionised water, erase, re-write with ethanol and erase, as shown in Fig. 4d. We also examined the time-dependent retention of the programmed SPL and the results in Fig. 4e showed that the SPL value arising from the $P_r$ of water was well-maintained over 500 h without significant SPL degradation (Supplementary Fig. 21b).

The poor retention of the stored information in a ferroelectric device is mainly due to the depolarisation of ferroelectric dipoles with time[49,50]. Depolarisation in a thin ferroelectric layer becomes severe when an insulating interlayer is employed. The formation of a depolarisation field is inevitable owing to large differences between the capacitance of the ferroelectric layer and that of the dielectric interlayer[51]. Utilisation of P(VDF-TrFE) enabled us to avoid these issue owing to the high ratio of the two capacitances of P(VDF-TrFE) and PMMA (~0.5). The successful retention of the SPL for water was achieved after 100 h at a temperature lower than 60 °C – this is lower than the Curie temperature of PVDF-TrFE (~80 °C; Supplementary Fig. 22b). Our LIFS devices are stable as long as 500 h with at least five different liquids (Supplementary Fig. 22a). We examined the morphologies of the PVDF-TrFE film before and after the sensing and memory operation (Supplementary Fig. 23). Characteristic needle-like crystalline domains are apparent with a length of ~400 nm, consistent with those observed in the previous studies[52,53]. The crystal orientation of a PVDF-TrFE film was also revealed with a grazing incident X-ray scattering technique (Supplementary Fig. 24), and results indicated that the c-axis of the crystal, i.e., the chain axis, was dominantly aligned parallel to the surface and perpendicular to the vertical electric field, consistent with those reported previously[53,54].

**Liquid position detection with a pad type LIFS.** The remnant-polarisation-dependent SPL of the ferroelectric layer programmed with a liquid allowed us to develop a position detection pad of a liquid droplet (Fig. 5). To demonstrate the proof-of-concept of our liquid detection pad, we fabricated four sets of 3 × 3 liquid

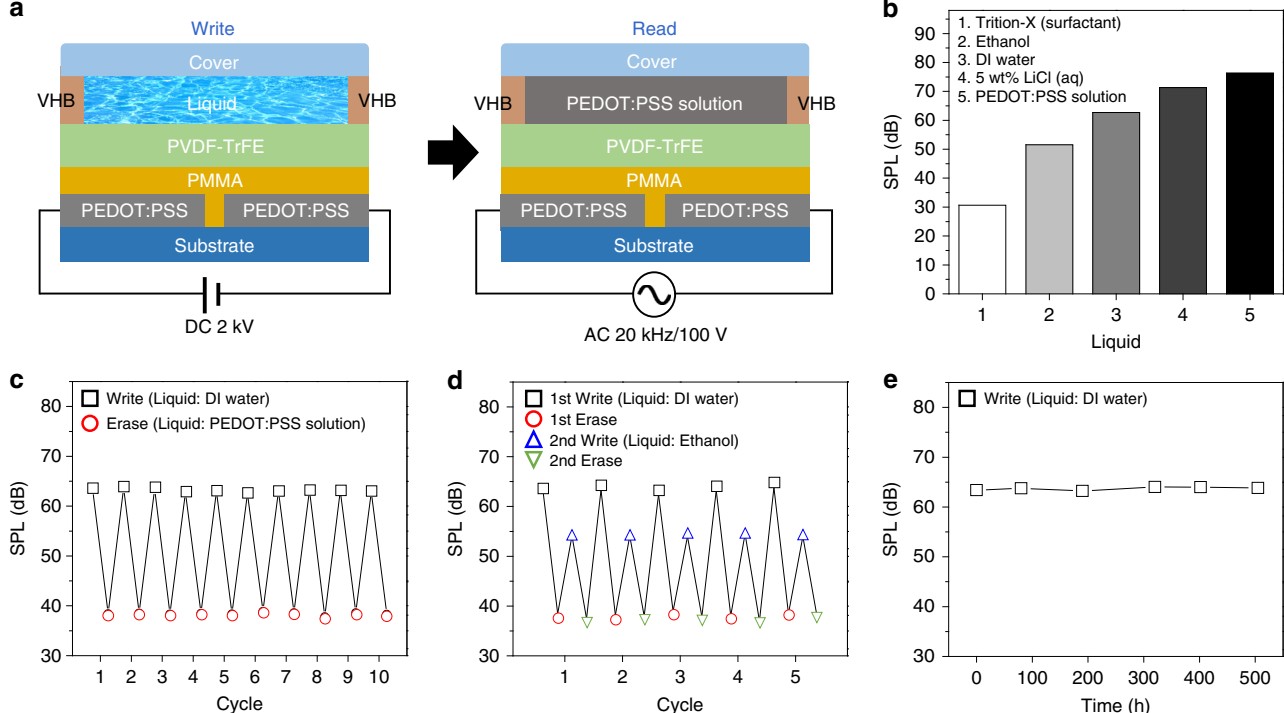

**Fig. 4** Non-volatile liquid-interactive sensing memory. **a** Schematics of the device structure for liquid sensing memory. An liquid-interactive ferroelectric sound (LIFS) arising from remnant polarisation of a liquid ($P_{r\text{-}L}$) developed in the write step is read in terms of SPL with a reference PEDOT:PSS solution in water. **b** Sound pressure level (SPL) values of LIFSs written with five liquids with different polarities and subsequently read with the reference PEDOT:PSS solutions after removal of the liquids. **c** Write-erase cycle endurance of an LIFS alternating current (AC) device with deionised water. SPL value arising from the LIFS of deionised water was substantially reduced after erasing the written remnant polarisation. Re-writing of the device with deionised water resulted in an SPL value almost identical to that in the first writing step. Ten write-erase cycles were reliably achieved with significant variation of SPL. **d** Variation of SPL values of an LIFS AC device upon multiple cycles of consecutive write with deionised water, erase, re-write with ethanol, and erase. Two characteristic SPL values of 65 and 55 dB written with deionised water and ethanol, respectively, are apparent after the multiple cycles. **e** Time-dependent retention of SPL value arising from an LIFS of deionised water. Direct current (DC) voltages of 2 and −1 kV were used for writing and erasing for all the liquids, respectively. The reading was obtained for all the liquids at an AC frequency and voltage of 20 kHz and 100 V, respectively. The electrode spacing of all samples is 12 mm

position detection pads for single and multi-droplet position detection. The four sets of pad devices with nine position-dependent SPL values were programmed for four different liquids, i.e., DI water, ethanol, a PEDOT solution and an LiCl solution, as shown in Fig. 5. Each 3 × 3 array pad was pro-grammed with nine different DC voltages ranging from 1.0 to 1.40 kV to develop nine different marking spots with different remnant polarisation values. As the SPL values arising from the nine different remnant polarisation values were also dependent upon the AC frequency (Fig. 2e), four different AC frequencies of 14, 16, 18, and 20 kHz were applied to the four zones 1, 2, 3, and 4, respectively, upon the reading process as shown in Fig. 5c. For instance, when a water droplet with a constant volume was deposited on one of the 3 × 3 spots of zone 2, a characteristic SPL with a maximum amplitude at 16 kHz was obtained depending on the position of the droplet, as shown in Fig. 5c. On the other hand, when an ethanol droplet with a constant volume was deposited on one of the 3 × 3 spots of zone 1, a characteristic SPL with a maximum amplitude at 14 kHz was obtained depending on the position of the droplet, as shown in Fig. 5c. Four sets of optically transparent, 3 × 3 arrays of LIFSs programmed and read by four different liquids were successfully developed as shown in Fig. 5d. Multi-droplet sensing was achieved when four droplets of four different liquids were deposited on the corresponding pads, followed by the different frequency reading as shown in Fig. 5e. The four droplets of ethanol (1–2), DI water (2–4), 5 wt% LiCl (a.q.) (3–6), and PEDOT:PSS solution (4–8) gave rise to the

characteristic SPL values with different maximum amplitudes of 14, 16, 18, and 20 kHz, as shown in Fig. 5e. Again, note that we were not able to detect two droplets of the same liquid on a 3 × 3 array device.

In order to detect multiple droplets of a liquid on a single 2D pad device, we simply designed 2 × 2 arrays of LIFSs with 4 AC generators (Supplementary Fig. 25a). When we again employed the 4 reading frequencies of 14, 16, 18, and 20 kHz right after one step polarisation poling programming instead of 9 different poling programming steps in Figs. 4 and 5 different SPL in each frequency were independently obtained without interference, allowing for the detection of all the combinations of a PEDOT: PSS solution of single, two, three, and four droplets on the pad device (Supplementary Fig. 25). For wearable or on-body applications, we believe that the sound of a single liquid from an LIFS can be readily detected with the microphone in a commercial cell-phone located at a fixed position from an LIFS, as conceptually illustrated in Supplementary Fig. 26.

**Dynamic liquid sensing and monitoring by LIFS.** Our LIFS AC device was advantageous for gaining dynamic information such as the velocity of a liquid flowing through confined channels with different dimensions; the results are shown in Fig. 6. The velocity of a liquid passing through a fluidic channel was measured by injecting the liquid into the fluidic channel at a known feeding speed and volume on the LIFS AC devices with three in-plane PEDOT:PSS electrodes, as shown in Fig. 6a. We designed the

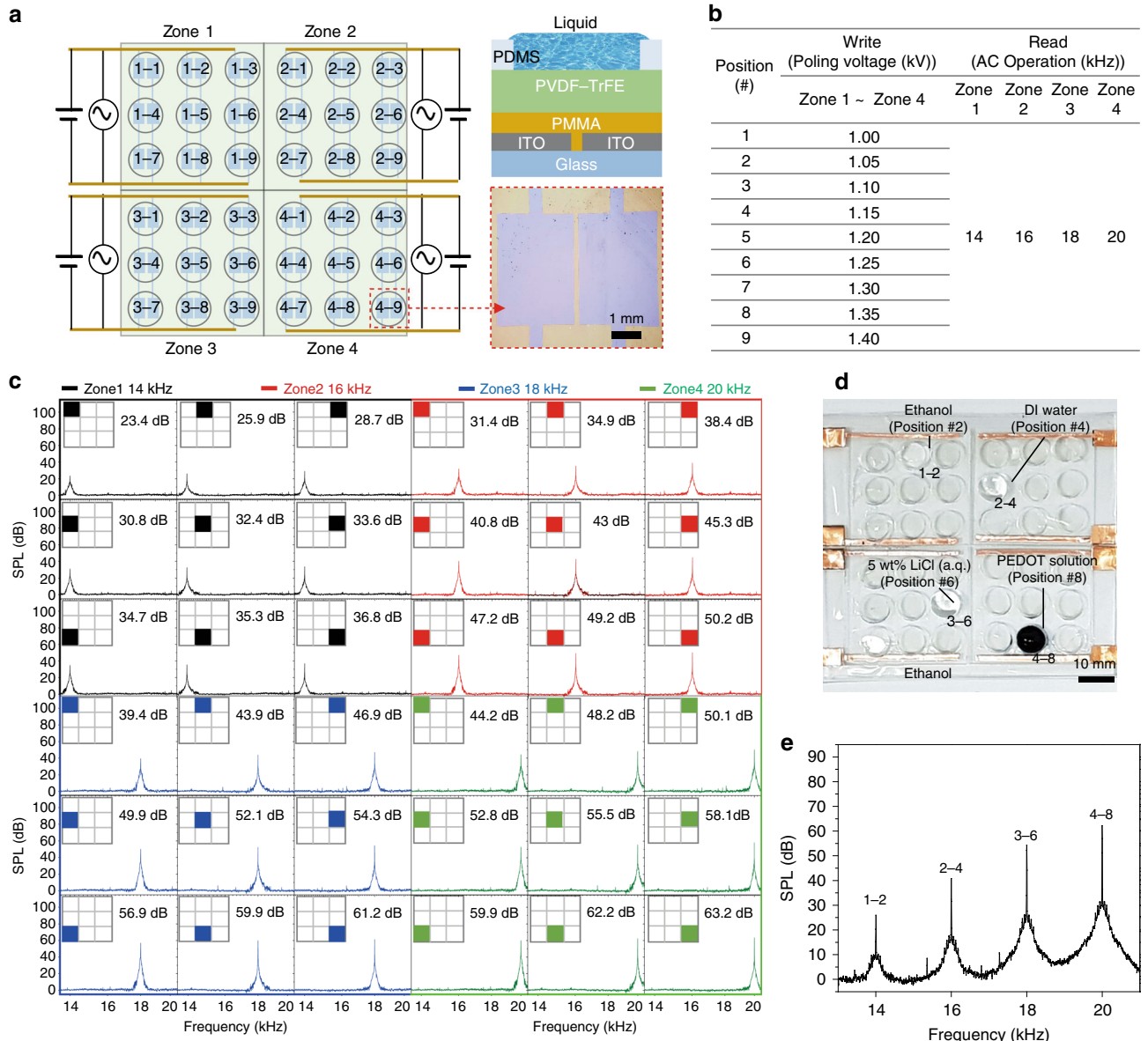

**Fig. 5** Position detection of a liquid droplet by liquid-interactive ferroelectric sound. **a** Schematics of four sets of 3 × 3 liquid position detection pads for single and multi-droplet position detection. A schematic and a photograph of a single liquid-interactive ferroelectric sound (LIFS) alternating current (AC) device pixel with PDMS space. **b** Table showing the position marking process. Each 3 × 3 array pad was programmed with nine different direct current (DC) voltages ranging from 1.0 kV to 1.40 kV to develop nine different marking zones with different remnant polarisation values. Zones 1, 2, 3, and 4 were programmed with Ethanol, DI water, 5 wt% LiCl (a.q.) and PEDOT:PSS solution, respectively. The four different AC frequencies of 14, 16, 18, and 20 kHz were applied to zones 1, 2, 3, and 4, respectively, during the reading process. **c** Sound pressure level (SPL) spectra of the nine positions of each zone. Owing to the different reading frequency values of 14, 16, 18, and 20 kHz, nine positions of each 3 × 3 array pad were clearly resolved in the SPL. Nine different SPL values for each liquid were obtained, depending on the position, allowing for sound-based position detection of a liquid. **d** A photograph of the position detection pad with Ethanol (1–2), DI water (2–4), 5 wt% LiCl (a.q.) (3–6), and PEDOT:PSS solution droplets (4–8). **e** SPL spectra arising from the four droplets on the position detection pad. All SPL values were obtained at a voltage of 100 V

width of each PEDOT:PSS electrode as 15 mm with a gap of 2 mm. The device was operated at the AC frequency of 20 kHz and voltage of 100 V with the pumping rate of 0.05 mL/s. When the advancing water flow crossed over the first gap and reached the point A in Fig. 6a, SPL rapidly increased owing to the vertical field arising from the filled water. After a certain duration, an additional increase in SPL occurred when the flow reached the point B (2nd bridge), as shown in Fig. 6a. Based on the time duration between the first and second increases in SPL, which was ~2.5 s, we could evaluate the velocity of the water flow in the channel as 5.9 mm/s.

Capillary flows passing through micrometre-scale channels were also successfully monitored with our LIFS AC device and the results are shown in Fig. 6b. Micro-channels of width and height 7 and 3 μm, respectively, were fabricated by placing a micro-patterned poly(dimethyl siloxane) (PDMS) mould[55]. For mimicking the capillary flow of human blood through veins, we employed a human serum fluid. As soon as a serum droplet was deposited on the inlet of the channels, the fluid was injected into the channels by capillary force. When the fluid crossed over the gap, SPL abruptly increased, as shown in Fig. 6b. Moreover, multiple fluids were monitored with a single LIFS AC device. A

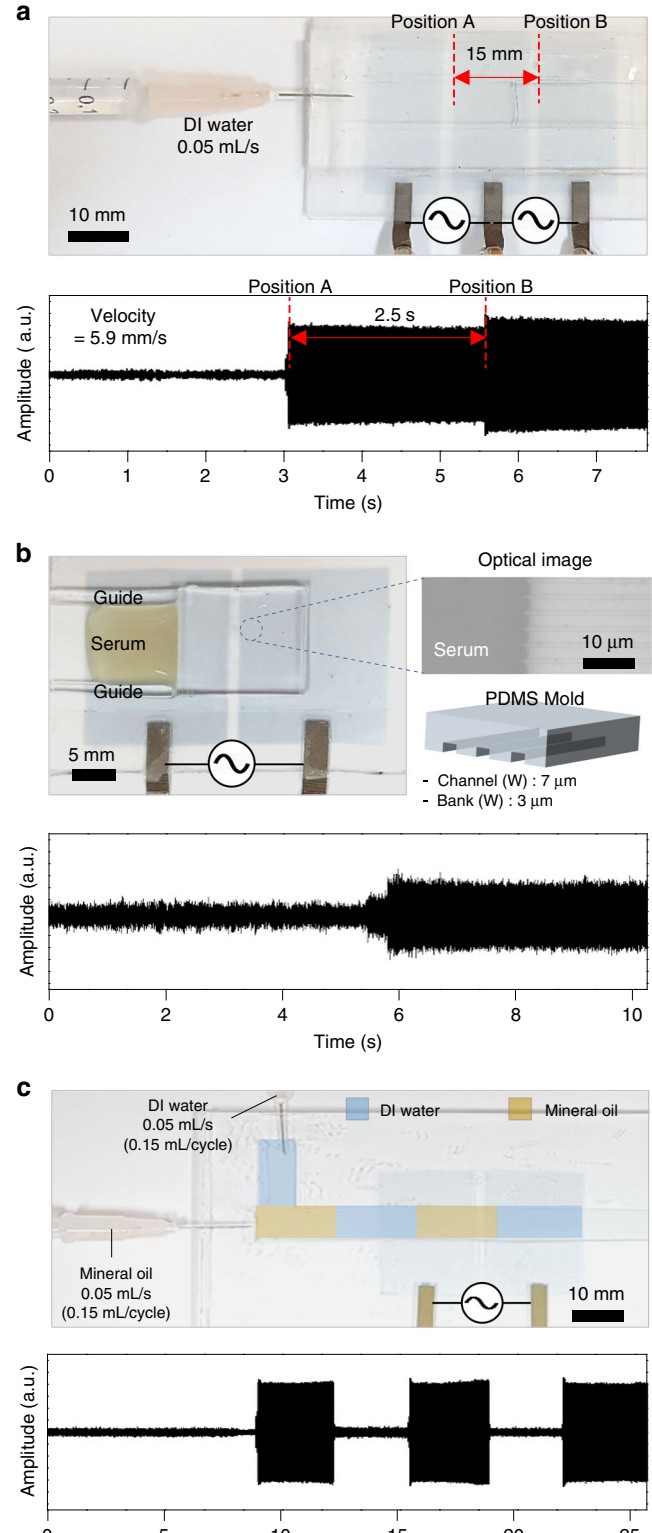

**Fig. 6** Dynamic monitoring of liquids by liquid-interactive ferroelectric sound. **a** A photograph of an liquid-interactive ferroelectric sound (LIFS) alternative current (AC) device with three in-plane PEDOT:PSS electrodes for monitoring the velocity of a liquid in microfluidic channels. Sound amplitude monitored with time when deionised water passed through the microfluidic channel. **b** A photograph of an LIFS AC device with two in-plane PEDOT:PSS electrodes for monitoring the flow of human serum in a capillary channel similar in size to the human capillaries. Sound amplitude monitored with time when the serum passed through the capillary. **c** A photograph of an LIFS AC device with two in-plane PEDOT:PSS electrodes for monitoring two different liquids in a microfluidic channel. Alternating flows of deionised water and mineral oil were developed with two injection systems. The characteristic sound amplitudes for deionised water and mineral oil were precisely monitored with time. All the devices were operated at the AC frequency and voltage of 20 kHz and 100 V, respectively

**Tube-type LIFS AC device with mechanical flexibility.** The tube-type liquid detection device with optical transparency is important, considering that most chemical and biological fluids are transported through cylindrical tubes including human blood systems. First, we examined the SPL sensing performance of our LIFS AC device, which consisted of transparent electrodes and a ferroelectric layer, as a function of bending radius, and the results are shown in Fig. 7a. For the flexibility test, we employed a gel-type hydrogel layer of thickness 0.5 mm as a top floating sensing material to eliminate possible errors arising from the deformed liquid during bending. The characteristic SPL value owing to the presence of the hydrogel layer was well-maintained within experimental uncertainty even after the device was bent with the bending radius of 2 mm, as shown in Fig. 7a. The excellent mechanical flexibility of the LIFS AC device allowed for the development of tube-type liquid sensing devices, as shown in Fig. 7b, c. A planar LIFS AC device was fabricated, and was subsequently rolled several times, resulting in a rolled LIFS AC device with a diameter of 2 mm. A sufficiently large SPL was obtained when deionised water was injected into the device with an AC frequency and voltage of 20 kHz and 100 V, respectively, as shown in Fig. 7c (Supplementary Movie 3).

More conveniently, a tube-type LIFS AC device was also developed with a pair of attachable in-plane Cu electrodes with a gap of 2 mm, as schematically shown in Fig. 7d. First, a tube of 50-μm-thick PVDF-TrFE/PMMA bilayer of diameter approximately 5 mm was connected with a syringe, followed by the gentle contact of the attachable Cu electrodes. When water injected from the syringe passed the second Cu electrode in contact with the surface of the PVDF-TrFE/PMMA tube, SPL abruptly increased owing to water-interactive ferroelectric actuation, as shown in Fig. 7e (Supplementary Movie 4). As the attachable Cu electrodes could be placed at any position of the tube, liquid detection in terms of SPL could be readily achieved on any area of the tube, as shown in Fig. 7e. The results indicate that our LIFS AC device could also be useful for the in situ analysis of a target liquid in the tube.

**Technological implication of an LIFS.** There are numerous acoustic based detection technologies representatively utilising the bulk acoustic wave (BAW) as well as surface acoustic wave (SAW). Both types of technologies based on a piezoelectric resonance frequency sensitive to the targeting elements placed on the layer have been widely used for sensing pressure, humidity, temperature, mass as well as a variety of chemical vapours and liquids[56–58]. Although the principle of devices based on BAW and SAW is different from that of our LIFS device, all of these

two-channel microfluidic platform was developed for demonstration as shown in Fig. 6c (Supplementary Fig. 27). Water and oils were sequentially injected through syringes, resulting in alternating oil/water fluid blocks with constant velocity, as schematically illustrated at the top of Fig. 6c. The movement of the alternating fluid blocks was successfully monitored in terms of SPL with time as shown at the bottom of Fig. 6c.

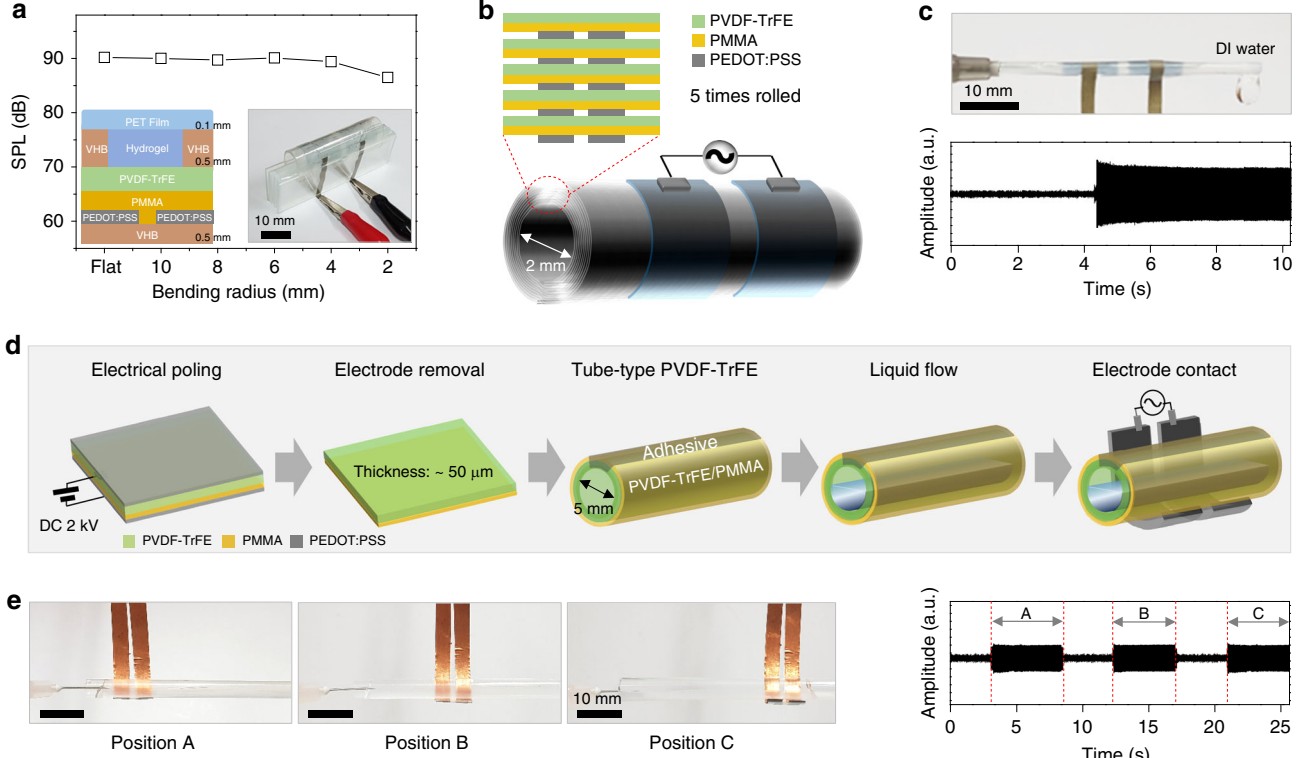

**Fig. 7** Mechanical flexible tube-type liquid-interactive ferroelectric sound device. **a** Sound pressure level (SPL) values of an liquid-interactive ferroelectric sound (LIFS) alternating current (AC) device as a function of the bending radius. Hydrogel was used as a liquid for the flexibility test. A photograph of a flexible device with the bending radius of 4 mm under measurement is shown in the inset. **b** Schematic of a rolled tube-type LIFS AC device. The device was fabricated by rolling a planar-type LIFS AC device five times, as shown in the schematics of the cross-sectional view of the device. The diameter and thickness of the tube were ~2 mm and 30 μm, respectively. **c** A photograph of the rolled tube-type LIFS AC device. Sound amplitude monitored with time when deionised water was passed through the tube. **d** Schematic of the fabrication of a tube-type LIFS AC device with attachable in-plane Cu electrodes. **e** Photographs of the tube-type LIFS AC device with attachable in-plane electrodes. The attachable electrodes facilitated in situ monitoring of a liquid at three different positions A, B, and C. Sound amplitude monitored with time at the positions A, B, and C of the photographs. All the devices were operated at the AC frequency and voltage of 20 kHz and 100 V, respectively

technologies can be categorised into sound-based detection systems. The utilisation of sound, rather than electric signals, can also be beneficial because sound does not require physical contact of the detection components and can readily propagate in space. This can allow for wireless detection of the liquid information and thus make it potentially suitable potentially for on and in-body applications, where a detection microphone can be remotely placed.

## Discussion
In summary, we demonstrated an AC operating platform capable of identifying, memorising, and monitoring the information of various liquids. Our sensing memory device utilised audible SPL arising from liquid-interactive ferroelectric actuation, which increased with the polarity of the liquids. The device worked with the characteristic LIFS resulting from a liquid droplet placed on a ferroelectric PVDF-TrFE layer across two in-plane PEDOT:PSS electrodes underneath the PVDF-TrFE/PMMA bilayer under an external in-plane AC field. More importantly, the LIFS of a liquid capable of being recorded in the form of non-volatile ferroelectric remnant polarisation allowed us to retrieve the information of the liquid even after its removal, resulting in a sensing memory of the liquid. By controlling the programming DC voltage and reading frequency, both of which were correlated with the SPL of the device, we demonstrated a thin film pad that could detect the local position of liquid droplets. Furthermore, the flow of a liquid

was monitored in terms of SPL when the liquid was injected into microfluidic channels on an LIFS AC device. Mechanically flexible and optically transparent tube-type LIFS AC devices were also developed, allowing for in situ monitoring of a liquid in terms of SPL. Our mechanically flexible and microfluidic compatible sensing memory based on LIFS can provide great potentials for a variety of biomedical diagnosis, hazardous liquid detection such as toxic solvents and volatile organic compounds (VOCs), health monitor/care devices including biomarker-free detection sensors, microfluidic cell counting and sorting SPL devices, and non-volatile sound-based touch pad for authentication.

## Methods
**Materials**. PVDF-TrFE was purchased from SOLVAY (250/P400). PEDOT:PSS (Clevios PH 1000), Zonyl surfactant (FS-300 fluoro-surfactant), and dimethyl sulfoxide (DMSO) were purchased from Sigma-Aldrich. Methyl ethyl ketone (MEK) was purchased from Sigma-Aldrich. PMMA was purchased from MICRO CHEM (950 PMMA A6). Acrylamide (AAm; Sigma, A8887), $N$, $N$-methylenebi-sacrylamide (MBAA; Sigma, M7279), $N,N,N',N'$-tetramethylethylenediamine (TEMED; Sigma, T7024), ammonium persulfate (AP; Sigma, A9164), and lithium chloride (LiCl; Sigma, L4408) were also purchased from Sigma-Aldrich and used to fabricate a hydrogel. All other chemicals including serum were purchased from Sigma-Aldrich and used as received. VHB 4905 and 4910 were purchased from 3M and used as received.

**Fabrication of the LIFS AC device**. The device architecture and fabrication process are illustrated in Supplementary Fig. 1. First, a Si wafer substrate was

sequentially cleaned with acetone and 2-propanol in an ultrasonic bath for 10 min each, followed by UV treatment for 15 min. PVDF-TrFE powders were dissolved in MEK solvent and stirred at 75 °C and 450 rpm for 2 h. The solution of PVDF-TrFE was spin-coated on the Si wafer substrate, followed by thermal annealing at 145 °C for 2 h in a glove box, yielding an ~5-μm-thick film. A thin PMMA film was spin-coated on the PVDF-TrFE layer, followed by thermal annealing at 140 °C for 1 h in ambient atmosphere, yielding an ~1.2-μm-thick film. The pristine PEDOT:PSS was modified by mixing it with 5 wt% DMSO and 0.5 wt% Zonyl surfactant with respect to PEDOT:PSS[59–61]. A PEDOT:PSS layer was thereafter spin-coated from the modified solution on the PMMA layer. An ~400-nm-thick PEDOT:PSS film was subsequently annealed at 110 °C for 1 h in ambient atmosphere. A gap was developed in the PEDOT:PSS layer through RIE ($O_2$ of 15 sccm, Ar of 3 sccm, time of 2.5 min) with a shadow mask, resulting in two in-plane PEDOT:PSS electrodes on the PMMA layer. The three layers PVDF-TrFE/PMMA/PEDOT:PSS were readily transferred onto another substrate such as flat glass, cylindrical surface, or human body. Device fabrication was completed when the two PEDOT:PSS electrodes were electrically connected as shown in Supplementary Fig. 1a. The planar-type LIFS device was developed by fabricating in-plane PEDOT:PSS electrodes on a glass substrate, followed by the sequential deposition of PMMA and PVDF-TrFE. VHB spacer was built on the device with cover to minimise possible evaporation of liquids during measurement as shown in Supplementary Fig. 1b. The detailed process conditions of fabricating an LIFS AC device are summarised in Supplementary Table 1.

**Molecular structure and polarisation of PVDF-TrFE film**. The molecular structures and microstructures of PVDF-TrFE layers were examined using high-resolution X-ray diffraction (HR-XRD) (Smart Lab). The piezoelectric polarisation of the PVDF-TrFE films was characterised with RT66.

**Electric field simulation of an LIFS AC device**. Numerical simulations of electrostatic interactions in the device were carried by open-source Finite element analysis (FEA) software Elmer (CSC – IT CENTER FOR SCIENCE). In the simulation, the dielectric constants of PVDF-TrFE and PMMA were 16 and 5, respectively. The voltage across two electrodes was 100 V. For floating conductor, constant potential boundary condition was applied.

**Piezoresponse force microscopy of PVDF-TrFE film**. The polarisation switching behaviour of PVDF-TrFE thin films was investigated with a commercial AFM (Multimode SPM, Bruker) equipped with PFM mode. A commercially available Cr/Pt-coated conductive coating AFM tip with a force constant of $3\,N\,m^{-1}$ and a resonance frequency of 75 kHz was used for the PFM measurements. The sample was mounted on a standard conductive holder disk, attached, and contacted by copper tape. The tip was grounded and a voltage was applied to the sample for the measurement of polarisation switching properties. The entire region was imaged with an AC tip bias (2.3 V), while the sample was grounded.

**Characterisation methods**. The cross-sectional and surface morphologies of the LIFS AC device were characterised using a field-emission scanning electron microscope (FESEM) (JEOL-7800F). The surface morphology of the PVDF-TrFE layer was analysed using tapping-mode AFM (Nanoscope Iva, Digital Instruments) in height and phase contrast. 2D grazing incidence X-ray diffraction experiments were performed at the Pohang Accelerator Laboratory in Korea. The films were placed on an x- and y-axis goniometer and were irradiated with monochromatized X-rays ($\lambda = 0.10608\,nm$) having grazing incidence angles ranging from 0.09 to 0.15°. A function generator (Agilent 33220 A) connected with a voltage amplifier (TREX 623B) was used for AC operation and electric field poling of the LIFS AC device. To evaluate the SPLs arising from ferroelectric actuation of a device, an omnidirectional microphone (Earthwork M23) was used and positioned at a distance of 100 mm from the LIFS AC device. The output sound signals produced from the LIFS AC device were converted to digital sound signals, and amplified using an audio interface (TASCAM US 2 × 2). The digital sound signals with waveforms in the time domain were converted to those in the frequency domain through fast Fourier transform (FFT) of a spectrum analyser (SIA Smart live) of a desktop computer. The induced voltage was measured by an oscilloscope (DPO 2022B, Tektronix, USA). The frequency-dependent SPL was measured in the frequency range 20 Hz to 20 kHz, which covers the range audible to the human ear (Supplementary Fig. 28). To minimise the effect of external noise, we selected the frequency domain of 20 kHz in most experiments and removed the signals of other ranges using a high-pass filter in FFT conversion.

## Data availability
The data that support the findings of this study are available from the authors on reasonable request, see author contributions for specific data sets.

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

## Acknowledgements

This study was supported by the Creative Materials Discovery Program through the National Research Foundation of Korea (NRF) funded by the Ministry of Science and ICT (2018M3D1A1058536). This study was also supported by a grant from the National Research Foundation of Korea (NRF) funded by the Korean government (MEST) (No. 2016M3A7B4910530 and 2017R1A2A1A05001160).

## Author contributions

J.S.K. and E.H.K. contributed equally to this work. Chanho P. and J.S.K. conceived and designed the experiments. J.S.K. performed the fabrication and device characterisation of the LIFS AC device. E.H.K., B.J. and K.L.K. performed the demonstration on the LIFS AC device and prepared the figures. S.W.L., I.H., H.H., S.L. and W.S. prepared the materials and performed measurements. G.K. and J.H. analysed the data. Cheolmin P. supervised the project and wrote the paper. All authors discussed the results and commented on the manuscript.

## Additional information

**Competing interests:** The authors declare no competing interests.

