## [Peer Review File · Nature Communications]

Reviewers' comments:

Reviewer #1 (Remarks to the Author):

This paper proposes a ferroelectric sound based liquid sensor and position detector based on PEDOT:PSS electrode and PVDF-TrFE ferroelectric materials. According to the polarity property of specific liquid against PVDF-TrFE, the corresponding polarization profile can be "written" into ferroelectric materials under a certain electric field. Based on the AC induced acoustic wave from this ferroelectric memory, the information can be "read" out through the sound pressure difference with respect to the reference site. Hence, the various types of liquids can be differentiated. A position detection array was also fabricated to with this mechanism, by integrating multiple PEDOT:PSS electrode lines under a single PVDF-TrFE ferroelectric materials. According to the comments provided below, this reviewer does not recommend acceptance of the manuscript in this stage and suggest major revision of the manuscript.

1. Ferroelectric based memory (FRAM) is a promising technology that is commercialized. This non-volatile memory has the advantage of long lifetime of writing, but limited lifetime of reading (losing its non-volatile property). Since the memorization mechanism of the proposed sensor is similar, can you comment on this issue according to your device?
2. Some of the liquids may have similar polarity index, such as ethanol and ethyl acetate, or methanol and acetone. How is the sensitivity and selectivity of LIFS in terms of sensing these liquids?
3. Since the proposed device only measured the dielectric constant (polarity) of the liquid, how about the mixture of different solutions? Normally, there is no pure solution in biomedical applications. The author demonstrates the condition for LiCl solution with different concentrations. However, I do not see how to identify the solution when the ingredients are unknown.
4. As shown in Figure 3a, the ferroelectric material PVDF-TrFE is directly contacted with liquid in order to alter the polarization profile. How is the stability after long term use, especially for corrosive liquids detection?
5. Authors showed that the retention time of SPL value arising from an LIFS of deionized water can be as long as 500 h. What is the environmental conditions of this measurement? Any parameters will affect the retention time? For other liquids?
6. The ferroelectric polarisation is determined by the exact electric field. And the exact electric field is determined by the equivalent capacitance of the liquid layer. This capacitance is calculated by the area, thickness and the dielectric constant of the liquid layer. So the dielectric constant is not the exclusive parameter to affect the output. Only when the other parameters are identical, the result can be consistent. However, for the test in Figure 4, such parameter control cannot be realized for a droplet. How can a consistent result to be achieved?
7. There are some typos need to be corrected: In the introduction part, Solvochromic should be Solvatochromic. In page 5 when talking about the measured voltage, 1(b) should be 1(c). In page 7, Optimisation should be optimization. In Fig. 4d, please double check the inserted values of SPL.

Reviewer #2 (Remarks to the Author):

This work demonstrated the liquid sensing through the "polarity-interactive ferroelectric sound". The results are interesting and useful for potential application in microfluidic and biomedical fields. However, the fundamental working mechanism of the devices demonstrated in this work is not clear. The author only illustrated the experimental phenomenon and provide the basic assumption for explaining the results. Lot of fundamental characterizations including the polarization distribution, electrical field distribution, sound wave spectrum and so on, need to be further carried out for verify the assumptions. For example, the description of the polarization state of the PVDF-TrFE film under different setup of AC electrical fields, and the explanation on the changes of sound with the injection of liquid are inappropriate for the present from. Therefore, I think that the paper could not be published on Nature Communication.

The detailed comments are listed as following:

1. In Figure 1 and Supplementary Figure 9, the author illustrated the polarization state of the ferroelectric films under the AC field applied with in-plane directions. However, the description is confusing.

(1) What is the role that PMMA played from the electrical point of view? As we know that PMMA is a dielectric material which is insulate, so the introduction of PMMA between the electrodes and the ferroelectric materials will no doubt change the electrical field distribution inside the planar structure when the electrical field was applied between the electrodes. The electrical field will mainly distributed inside the PMMA layer but not the ferroelectric layer. As a result, the poling behavior of the PVDF-TrFE films may be changed.

(2) The DC or AC voltage was applied by the adjacent electrodes with in-plane configuration. However, the author claimed that the electrical field was applied along the vertical direction due to the existence of top electrode (in solid or liquid form). That conclusion is hard to be understood. Although Figure 2a in the supplementary materials shows an induced AC voltage signal with a half amplitude and the same phase of the input signal, that result came from the electrical potential difference (the completely same phase of the voltage signal could be a direct evidence because there should be a delay due to the capacitance behavior of the PVDF and PMMA layer) but can not be the evidence of an external electrical field applied along the vertical direction. So the applied electrical field as well as the poling direction of the PVDF-TrFE film should be along the in-plane direction, so does the deformation of the PVDF-TrFE film if we assume that the film was working at 33 mode. Theoretical simulation with Finite element analysis method and experimental measurement of in-plane and out-of-plane polarization by using the piezoresponse force microscopy could be helpful and essential for the authors to understand the electrical field distribution and the polarization direction of the PVDF-TrFE films.

2. The author should provide the sound wave spectra to confirm whether the variation of sound pressure level came from the amplitude changes or the frequency shift of the sound wave, because the sensitivity of the microphones and human ears always depends on the frequency of the sound.

3. Figure 3 in supplementary materials shows the angle-dependent SPL of the LIFS device. Please give detailed explanation of this interesting phenomenon.

4. The author compared the SPL level of liquid with different polarity or LiCl solution with different concentration. However, as a sensor, those qualitative results are not sufficient. The author should find out the quantitative relationship between the SPL level and a certain physical parameter of the liquid, e.g. the relative permittivity, pH value or viscosity, density and so on. So we can understand the fundamental mechanism by building a quantitative physical model for the phenomenon and performance of devices. From the sensor point of view, a quantitative calibration curve is also essential for practical application. Therefore, I strongly recommend that the author could find out the fundamental and quantitative relationship between some of the physical parameters of the liquid and the SPL level of the devices.

5. As known, the film bulk acoustic resonator (FBAR) based on the ferroelectric/piezoelectric thin films could also be used as a liquid sensor due to the shift of resonance frequency and amplitude induced by the change of mass loading on the top surface of the piezoelectric layer. Is there any possibility that the LIFS device demonstrated in this work are based on the mass-induced modulation of the resonance behavior of the PVDF-TrFE film? Because the results are so similar with the FBAR based devices. In addition, what is the resonance frequency of the PVDF-TrFE film in the LIFS device? The vibration of the PVDF-TrFE film could be largely amplified if the film was working at its resonance frequency.

6. Please check the words "on/off ratio" and "on/off difference" in the text, is there any difference? Or is it just spelling mistake?

7. What is the morphology and crystal structure of the PVDF-TrFE film? The authors should provide experimental data to support their conclusions on the composition, morphology, crystal structure and thickness of the materials.

8. PFM or macroscale piezoelectric response testing should be done for measuring the vibration state of the PVDF-TrFE films with either vertically or laterally applied AC and DC voltage, which could provide useful information for the explanation of the working mechanism of the device.

Reviewer #3 (Remarks to the Author):

In their manuscript, J. S. Kim and co-authors report methods and devices for sensing liquids with polarity interactive ferroelectric sound. Based on solution-based manufacturing processes, the authors create functioning devices and perform electro-acoustic characterization measurements. They use various device architectures and implementations to show how their devices perform in basic modes of operation. This includes differentiating and memorizing liquids based on their polarity, as well as monitoring flow dynamics.

In the current version of the manuscript, it is unclear to what extent the presented results constitute a scientific discovery. The authors have to clearly state what is new and establish a better context with the existing literature. Technologically, while a need for integrated liquid sensing technology indeed exists, it is unclear to what extent the method discussed here can be applied to solve actual problems.

In order for the manuscript to be considered for publication in Nature Communications, the authors should consider the questions and comments provided below and clearly state what is scientifically new while substantiating the technological relevance of their work:

- Fundamentals of LIFS: There are no literature references to fundamentals of liquid-interactive ferroelectric actuation and sound generation, as well as electro-acoustical liquid-solid interactions in the introductory paragraph. The authors should include suitable references here.
- Can the measured, vertical ac voltage or the sound amplitude be modeled based on the physical properties of the materials involved, taking into consideration the actual device geometry? The manuscript does not present or refer to any theoretical predictions or model simulations of expected device performance.
- Differentiation between liquids: Could the authors explain how the measured electrical or sound signal enables a unique and unambiguous identification of the chemical identities of the tested liquids?
- Considering potential applications in microfluidics, can the authors comment on the ultimate spatial resolution that can be achieved with their method; for example, in the measurement of the 2-D position of a liquid droplet (minimum droplet volume?) on a thin pad-type device? Integration and scaling properties of the technology should be discussed.
- What is the benefit of using the sound from the ferroelectric vibration to identify the liquids, instead of using the vertical ac voltage? As the sound signal decreases with electrode size, this could severely limit the miniaturization and integration of sensors reported here. Are the authors considering integration of local sound sensors in order to apply and scale within a technological context? The applicability of the method in a biomedical context seems limited due to the reliance

on an external microphone.

- The authors state on page 11 of the main manuscript that conventional, physical pixel arrays are not required in their method. How is the sound signal from individual sensors generated, identified and differentiated in case multiple sensors are used simultaneously and need to be read out in parallel, for example, in the case of a pad-type device sensing a two-phase liquid (oil/water) where spatial differentiation between multiple liquids/droplets is required?

- Can the authors explain how the result in Supplementary Fig. 2d relates to Fig. 1 c of the main manuscript that demonstrates a voltage drop for DI water in the same frequency range?

- For improving clarity, the authors should clearly identify and label the voltages discussed in text and figures. For example, which voltage is plotted on the horizontal axis of Supplementary Fig. 6 a?

- The authors should plot sound pressure level values versus polarity values for liquids in Fig. 3 b to support the statement on page 9 of the main manuscript that those values are proportional.

- How large are performance variations from device to device, for example in the measurement shown in Fig. 1 c ? The authors should consider including statistical data for evaluation of technological relevance.

Pointwise responses to the comments of the reviewers are appended below. We hope that the modified version of the manuscript will be acceptable for publication in *Nature Communications*.

Reviewer #1 (Remarks to the Author):

This paper proposes a ferroelectric sound based liquid sensor and position detector based on PEDOT:PSS electrode and PVDF-TrFE ferroelectric materials. According to the polarity property of specific liquid against PVDF-TrFE, the corresponding polarization profile can be “written” into ferroelectric materials under a certain electric field. Based on the AC induced acoustic wave from this ferroelectric memory, the information can be “read” out through the sound pressure difference with respect to the reference site. Hence, the various types of liquids can be differentiated. A position detection array was also fabricated to with this mechanism, by integrating multiple PEDOT:PSS electrode lines under a single PVDF-TrFE ferroelectric materials. According to the comments provided below, this reviewer does not recommend acceptance of the manuscript in this stage and suggest major revision of the manuscript.

1. Ferroelectric based memory (FRAM) is a promising technology that is commercialized. This non-volatile memory has the advantage of long lifetime of writing, but limited lifetime of reading (losing its non-volatile property). Since the memorization mechanism of the proposed sensor is similar, can you comment on this issue according to your device?

Response: We appreciate the reviewer’s comments. As the reviewer pointed out, as our LIFS device also utilizes ferroelectric polarization for storing and reading the information of polar liquids, the limitations of devices containing ferroelectric materials should be carefully considered. We have systematically studied the retention properties of non-volatile ferroelectric memories containing PVDF-TrFE layers in the device architectures of a capacitor and a field effect transistor [*Appl. Phys. Lett.* **92**, 012921 (2008), *Adv. Funct. Mater.* **19**, 1609–1616 (2009)]. The poor retention of the stored information in a ferroelectric device is mainly due to the depolarization of ferroelectric dipoles with time. In particular, depolarization in a thin ferroelectric layer becomes severe when an insulating interlayer is employed. The theoretical work by Ma *et al.* (*IEEE Electron Device Lett.* **23**, 386 (2002)) has suggested that the depolarization field is inevitable due to the large difference between the capacitance of the ferroelectric layer (C_F) and that of the dielectric interlayer (C_I). In particular, the ratio of C_F to C_I is inversely proportional to the depolarization field (E_{dp}) according to the relation $E_{dp} = P [\varepsilon (C_I/ C_F + 1)]^{-1}$, where P and ε correspond to the polarization and dielectric constant of the interlayer, respectively. A typical value of the ratio in an inorganic ferroelectric memory, for example, with PZT and SiO₂ is smaller than 0.005, leading to very poor data retention. Utilization of P(VDF-TrFE) enabled us to avoid these issues due to the high ratio of the two capacitances of P(VDF-TrFE) and PMMA (~0.5). The depolarization field of polymeric ferroelectrics such as PVDF-TrFE is negligibly influenced by the PMMA interlayer in our LIFS device; therefore, we are able to achieve long-term retention of the stored information of a liquid for over 500 h. We have explained this on pages 15 and 16 of the revised manuscript.

2. Some of the liquids may have similar polarity index, such as ethanol and ethyl acetate, or methanol and acetone. How is the sensitivity and selectivity of LIFS in terms of sensing these liquids?

Response: As our LIFS device is based on the polarity-dependent electric field upon AC application, it is not trivial to distinguish two liquids with similar polarities as described in the manuscript. Complementary characterization tools, such as FT-IR, HPLC, and NMR, should be employed in addition to sensing with an LIFS device to clearly identify the two liquids. To address the issues of the sensitivity of LIFS devices as a function of the polarity (or dielectric constant) of a solvent, we extensively examined 18 solvents with different polarities, and have provided the polarity-dependent SPL results in a new Fig. 2c and Supplementary Table 2. The results show that the SPL is almost linearly proportional to the dielectric constant values of the solvents, and the sensitivity of an LIFS device is approximately $0.75 \text{ (dB}/\epsilon_r)$, as obtained from the slope of the curve. We have added this explanation on page 10 of the revised manuscript.

Figure 2. (c) SPL values of 18 liquids examined with different dielectric constants, and the sensitivity of an LIFS AC device obtained from a linear slope of the curve. The device was operated at an AC frequency and voltage of 20 kHz and 100 V, respectively.

No	Solvent	Polarity index	Dielectric constant (ϵ_r)	SPL (dB)	PVDF-TrFE solvent
1	Cyclohexane	-0.2	2.02	0	
2	Mineral oil	0.0	2.30	0	
3	n-Hexane	0.1	1.88	0	
4	Trichloroethylene	1.0	3.40	0	
5	f-Propyl ether	2.4	3.90	2.03	
6	Chlorobenzene	2.7	5.60	10.32	
7	Octoxynol-9	3.2	7.50	15.31	
8	n-Octanol	3.4	10.30	20.22	
9	n-butanol	3.9	17.50	29.23	
10	Iso-propanol	4.1	20.30	32.22	
11	Ethanol	4.3	24.60	34.52	
12	Benzonitrile	4.8	25.20	35.01	
13	Methanol	5.1	32.70	38.50	
14	Acetonitrile	5.8	37.50	40.85	
15	Dimethylformamide	6.4	36.70	-	○
16	Ethylene glycol	6.9	37.70	41.24	
17	Dimethylsulfoxide	7.2	47.00	-	○
18	DI water	10.2	78.20	66.54	

Supplementary Table 2. Characteristics of 18 solvents having different polarities, dielectric

constants, and polarity-dependent SPL values on an LIFS AC device.

3. Since the proposed device only measured the dielectric constant (polarity) of the liquid, how about the mixture of different solutions? Normally, there is no pure solution in biomedical applications. The author demonstrates the condition for LiCl solution with different concentrations. However, I do not see how to identify the solution when the ingredients are unknown.

Response: We appreciate the reviewer's comment on the sensing of a liquid mixture, especially one with an unknown ingredient. We have attempted to address this issue with our LIFS device; however, at this stage of the research, identification of the chemical structure of liquids and/or ingredients dissolved in a known liquid is hardly achievable because the SPL of our device only responds to the polarity of a liquid. Based on the master curve we obtained with a variety of liquids with different dielectric constant values (a new figure, Fig. 2c, and a new table, Supplementary Table 2), we will be able to predict the polarity information of an unknown liquid or a liquid with an unknown ingredient. Additional characterization should be performed for further identification. We hope that the reviewer understands that the development of a single sensing device that can identify an unknown ingredient dissolved in a solvent is quite challenging. We have briefly mentioned this issue on page 11 of the revised manuscript.

4. As shown in Figure 3a, the ferroelectric material PVDF-TrFE is directly contacted with liquid in order to alter the polarization profile. How is the stability after long term use, especially for corrosive liquids detection?

Response: We appreciate the insightful comments on the stability of liquid/ferroelectric interfaces with time. Liquids which solvate PVDF-TrFE cannot be suitable for liquid sensing. As we built the master curve of the dielectric constant-dependent SPL (new Fig. 2c and Supplementary Table 2), the solvents that solvate PVDF-TrFE, such as THF, MEK, pyridine, DMF, and DMSO, were excluded. The sensing of these solvents should be performed by employing a protective layer that is insoluble in the solvents. As a representative example, we placed a thin epoxy layer insoluble in DMF on top of the PVDF-TrFE layer of the LIFS device. The characteristic SPL values of DMF and DMSO were successfully obtained using this protective epoxy layer, as shown in a new Supplementary Fig. 10.

The long-term stability of the liquid/PVDF-TrFE interfaces was indirectly examined with the operation stability of LIFS devices containing various solvents. All the devices with five different liquids show long-term retention of information over 500 h, as shown in a new figure, Supplementary Fig. 22a.

Supplementary Figure 10. LIFS AC device with a protective layer for sensing a corrosive liquid. (a) Schematics and photographs of the device structure for the corrosive liquid sensing with and without a thin protective epoxy layer. No damage to the PVDF-TrFE layer with DMF was observed with a protective layer. (b) SPL values of five liquids including corrosive liquids such as DMF and DMSO examined with different dielectric constants on an LIFS AC device. The SPL values are proportional to the dielectric constants of the liquids. The device was operated at an AC frequency and voltage of 20 kHz and 100 V, respectively.

5. Authors showed that the retention time of SPL value arising from an LIFS of deionized water can be as long as 500 h. What is the environmental conditions of this measurement? Any parameters will affect the retention time? For other liquids?

Response: The retention was measured at ambient conditions with a relative humidity ranged from 30 to 60 % at room temperature. We also examined the effect of temperature on the retention of the SPL, and the results are shown in Supplementary Fig. 22b. As expected, the successful retention of the SPL for water was achieved after 100 h of measurement at a temperature lower than 60 °C; this is lower than the Curie temperature of PVDF-TrFE (~ 80 °C). In addition, the retention properties of LIFS devices with various liquids show that our LIFS devices are stable as long as 500 h with at least five different liquids in the new Supplementary Fig. 22a. We have explained this on page 16 of the revised manuscript.

Supplementary Figure 22. Stability of non-volatile LIFS AC memory device. (a) Variation in SPL values with time arising from five liquids with different polarities on an LIFS AC device. DC voltages of 2 and -1 kV were used for writing and erasing all the liquids, respectively. The reading was obtained for all the liquids at an AC frequency and voltage of 20 kHz and 100 V, respectively. (b) Characteristic SPL value after a 100-h retention arising from deionized water at different temperatures lower than the Curie temperature of PVDF-TrFE. The reading was obtained for the water at an AC frequency and voltage of 20 kHz and 100 V, respectively.

6. The ferroelectric polarisation is determined by the exact electric field. And the exact electric field is determined by the equivalent capacitance of the liquid layer. This capacitance is calculated by the area, thickness and the dielectric constant of the liquid layer. So the

dielectric constant is not the exclusive parameter to affect the output. Only when the other parameters are identical, the result can be consistent. However, for the test in Figure 4, such parameter control cannot be realized for a droplet. How can a consistent result to be achieved?

Response: We appreciate the reviewer's insightful comment. We fully agree with the reviewer that the ferroelectric polarization depends upon various parameters, such as the dielectric constant, area, and heights of a liquid. To address the issues the reviewer raised, firstly, we performed an electric field calculation based on the finite element method (FEM), with which we were able to confirm the vertical field arising from the top conductive layer (new Figs. 1d and 1e). A typical in-plane and fringe field developed between two in-plane electrodes on which PMMA and PVDF-TrFE layers were placed without a top conductive layer. When a top conductive layer was deposited, a vertically driven electric field developed on the two overlapped regions of the conductive layer with the two in-plane electrodes. Further experiments also revealed that the vertical field was conductive-dependent, as shown in a new Supplementary Fig. 4. We employed a top PEDOT:PSS layer whose conductivity was controlled by adding DMSO. The vertical field decreased with decreasing conductivity. We also observed that the vertical field was dependent upon the ion conductivity as well as the polarity of the top layer, and have explained this in the manuscript.

In our LIFS device with a vertical electric field, the vertical field distinctly depends on the overlapped area. The SPL values arising from a polar liquid changed when the contact areas were altered, as shown in the Supplementary Fig. 13. Interestingly, the electric field rarely changed when the heights of the fluid was varied, as shown in the new Supplementary Fig. 14. We evaluated the SPL of an LIFS device with different heights of deionized water. As described in the manuscript, the water was confined in the reservoir with a constant contact area, and the water height was varied. Almost identical SPL values were obtained, regardless of the height of the water in the range from a few micrometres to a few millimetres. The results imply that the ferroelectric polarization of our LIFS device is mainly governed by the interfacial properties of PVDF-TrFE/liquid, such as the contact area and dielectric properties of the liquid. To make our SPL results consistent, we set up a reservoir with a constant contact area as schematically shown in Fig. 2a. The constant contact area was also obtained in a tube type LIFS device. We have described this on pages 6,7,11,12 of the revised manuscript.

Figure 1. Architecture and working principle of the LIFS device. Finite element method (FEM) results of the LIFS AC device under a voltage bias between two in-plane electrodes showing both the direction and magnitude of the generated electric field without (d) and with (e) a top conductive layer on PVDF-TrFE.

Supplementary Figure 4. Dependence of the SPL and vertical AC voltage of the LIFS AC devices on the various properties of liquids. (a) SPL values and voltages induced by a vertical electric field arising from PEDOT:PSS with different conductivities controlled by adding DMSO on an LIFS AC device. (b) SPL values and voltages induced by a vertical electric field arising from six liquids with different dielectric constants on an LIFS AC device. (c) SPL values and voltages induced by a vertical electric field arising from different ionic conductivities controlled by the amount of LiCl in DI water on an LIFS AC device. All devices were operated at an AC frequency and voltage of 20 kHz and 100 V, respectively.

Supplementary Figure 13. SPL values arising from a polar liquid with different overlapped electrode areas on an LIFS AC device. The device was operated at an AC frequency and a voltage of 20 kHz and 100 V, respectively.

Supplementary Figure 14. SPL values of an LIFS AC device as a function of the heights of a polar liquid. (a) Schematic and photograph of an LIFS AC device with DI water, whose

volume can be controlled by the height of PDMS reservoir. SPL spectra (b) and values (c) arising from DI water with different heights on an LIFS AC device. The device was operated at an AC frequency and voltage of 20 kHz and 100 V, respectively.

7. There are some typos need to be corrected: In the introduction part, Solvochromic should be Solvatochromic. In page 5 when talking about the measured voltage, 1(b) should be 1(c). In page 7, Optimisation should be optimization. In Fig. 4d, please double check the inserted values of SPL.

Response: We thank the reviewer for bringing this to our attention. We have carefully checked and corrected the typos in the manuscript, including those the reviewer mentioned.

Reviewer #2 (Remarks to the Author):

This work demonstrated the liquid sensing through the “polarity-interactive ferroelectric sound”. The results are interesting and useful for potential application in microfluidic and biomedical fields. However, the fundamental working mechanism of the devices demonstrated in this work is not clear. The author only illustrated the experimental phenomenon and provide the basic assumption for explaining the results. Lot of fundamental characterizations including the polarization distribution, electrical field distribution, sound wave spectrum and so on, need to be further carried out for verify the assumptions. For example, the description of the polarization state of the PVDF-TrFE film under different setup of AC electrical fields, and the explanation on the changes of sound with the injection of liquid are inappropriate for the present from. Therefore, I think that the paper could not be published on Nature Communication.

The detailed comments are listed as following:

1. In Figure 1 and Supplementary Figure 9, the author illustrated the polarization state of the ferroelectric films under the AC field applied with in-plane directions. However, the description is confusing.

Response: We thank the reviewer for bringing this to our attention. We have improved the clarity of our description of the working principle of the device by explicitly addressing the issues the reviewer raised below.

(1) What is the role that PMMA played from the electrical point of view? As we know that PMMA is a dielectric material which is insulate, so the introduction of PMMA between the electrodes and the ferroelectric materials will no doubt change the electrical field distribution inside the planar structure when the electrical field was applied between the electrodes. The electrical field will mainly distributed inside the PMMA layer but not the ferroelectric layer. As a result, the poling behavior of the PVDF-TrFE films may be changed.

Response: We partly agree with the reviewer because it is true that a part of the vertical electric field was exerted on the PMMA layer. The performance of the device for sensing and memorizing the information of a liquid was, however, not significantly influenced as described below. To elucidate the electric field developed in our LIFS device, we first performed an electric field calculation based on the finite element method (FEM), with which we were able to confirm the vertical field arising from the top conductive layer; the results are shown in the new Fig. 1d and 1e. A typical in-plane and fringe field developed between two in-plane electrodes on which PMMA and a PVDF-TrFE layers were placed without a top conductive layer. When a top conductive layer was deposited, a vertically driven electric field developed on the two overlapped regions of the top conductive layer with the two in-plane electrodes. Notably, the vertical fields on the two overlapped areas were opposite in direction, which implies that the top conductive layer acted like an electric field mirror. Further experiments also revealed that the vertical field was conductive-dependent, as shown in a new figure, Supplementary Fig. 4. We employed a top PEDOT:PSS layer, whose conductivity was controlled by adding DMSO. The vertical field decreased with decreasing conductivity. We also observed that the vertical field was dependent on ion conductivity as well as the polarity of the top layer, as explicitly shown in the manuscript. It should be noted

that the polarity-dependent vertical electric field observed with various polar liquids could not be simulated with FEM owing to the complexity of the system.

In our LIFS device with the vertical electric field, two insulating layers of PVDF-TrFE and PMMA are connected in series between one of the two in-plane electrodes and a liquid. Note that the ferroelectric PVDF-TrFE layer is also an insulator. The arrangement is equivalent to that of two capacitors connected in series. As the reviewer mentioned, the electric field is shared with the two insulating layers according to a simple relation. The portion of the electric field exerted on either the PVDF-TrFE or PMMA layer depends on the dielectric constants and thicknesses of the layers, as shown in the equation. Based on the dielectric constants and the thickness of the PVDF-TrFE (5 μm) and PMMA (1.2 μm) layers, we were able to calculate the electric field exerted on the PMMA layer, which was approximately 30 % of the applied electric field. The electric field distribution of the device with a PMMA/PVDF-TrFE bilayer clearly shows the vertical electric field in the PVDF-TrFE layer, although the field strength is somewhat reduced owing to the presence of the PMMA layer, as shown in Supplementary Fig. 5. In our experiments, the electric field exerted on the PVDF-TrFE layer was large enough for the ferroelectric polarization switching needed for sensing and memorizing the information of a liquid. We have added this explanation on pages 6 and 7 of the revised manuscript.

Figure 1. Architecture and working principle of the LIFS device. Finite element method (FEM) results of the LIFS AC device under a voltage bias between two in-plane electrodes showing both the direction and magnitude of the generated electric field without (d) and with (e) a top conductive layer on PVDF-TrFE.

Supplementary Figure 4. Dependence of the SPL and vertical AC voltage of the LIFS AC devices on various properties of liquids. (a) SPL values and voltages induced by a vertical electric field arising from PEDOT:PSS with different conductivities controlled by adding DMSO to the LIFS AC device. (b) SPL values and voltages induced by a vertical electric field arising from six liquids with different dielectric constants on an LIFS AC device. (c) SPL values and voltages induced by a vertical electric field arising from different ionic conductivities controlled by the amount of LiCl in DI water on the LIFS AC device. All devices were operated at an AC frequency and voltage of 20 kHz and 100 V, respectively.

Supplementary Figure 5. Finite element method (FEM) results of the LIFS AC device with (a) PVDF-TrFE layer and (b) PMMA/PVDF-TrFE bilayer under a voltage bias between two in-plane electrodes showing both the direction and magnitude of the generated electric field with a top conductive layer on PVDF-TrFE.

(2) The DC or AC voltage was applied by the adjacent electrodes with in-plane configuration. However, the author claimed that the electrical field was applied along the vertical direction due to the existence of top electrode (in solid or liquid form). That conclusion is hard to be understood. Although Figure 2a in the supplementary materials shows an induced AC voltage signal with a half amplitude and the same phase of the input signal, that result came from the electrical potential difference (the completely same phase of the voltage signal could be a direct evidence because there should be a delay due to the capacitance behavior of the PVDF and PMMA layer) but can not be the evidence of an external electrical field applied along the vertical direction. So the applied electrical field as well as the poling direction of the PVDF-TrFE film should be along the in-plane direction, so does the deformation of the PVDF-TrFE film if we assume that the film was working at 33 mode.

Theoretical simulation with Finite element analysis method and experimental measurement of in-plane and out-of-plane polarization by using the piezoresponse force microscopy could be helpful and essential for the authors to understand the electrical field distribution and the polarization direction of the PVDF-TrFE films.

Response: One of the main goals of our work was to develop and utilize (sense and memorize) a vertical AC electric field between one of two in-plane electrodes, between which an in-plane AC field had been applied, and either a polar liquid or solid surface. It is unfortunate that the description of the development of the vertical AC field in the current version of the manuscript is not convincing enough. Although the vertical electric field arising from the in-plane AC field was evident and successfully utilized for novel field-

induced displays [*Nat. Commun.* **8**, 14964-14971 (2017) and *Adv. Mater.* **29**, 1703552 (2017)], we employed the finite element analysis method (FEM) to confirm the development of the vertical electric field of our LIFS device, as the reviewer suggested. The results shown in the new figures Figs. 1d and 1e clearly reveal that the vertical electric field between a conductor and one of the electrodes was developed on the area overlapped with the electrode. It is worth noting that the vertical fields on the two overlapped areas were opposite in direction, which implies that the top conductive layer acted as an electric field mirror. The experimental results also show that the vertical field developed in the overlapped area depends on not only on the conductance but also the polarity of the top layer (Supplementary Fig. 4 and 13 in the revised manuscript). We also described the polarity-dependent vertical electric field exerted on both the PMMA and PVDF-TrFE layers with a simple linear electric field model, as shown in Supplementary Fig. 6. The d_{33} of a PVDF-TrFE layer was aligned vertically and thus responsible for the ferroelectric polarization which depended on the polarity of liquids, as experimentally shown in the manuscript. The vertical alignment of the d_{33} mode of a PVDF-TrFE film in our LIFS device was additionally confirmed with a piezoresponse microscope, and the results are shown in a new figure, Fig. 3. PFM images in the piezoelectric amplitude contrast show that the piezoelectric amplitude significantly varies on the two overlapped regions, compared with the gap area. On the other hand, no piezoelectric response is observed on the PVDF-TrFE film without a top conductive layer when an in-plane field was applied. As expected from the FEM results in which the field directions of the two overlapped areas were opposite, the piezoelectric response in the two overlapped regions is opposite in amplitude, as evidenced in the amplitude profile with the schematic ferroelectric dipoles in the corresponding regions in Fig. 3c and 3d. As the vertical field developed in the overlapped area is polarity-dependent, the maximum piezoelectric amplitude is also polarity-dependent. The results shown in Fig. 3e clearly demonstrate that, the more polar a liquid, the higher the amplitude that develops, which is consistent with the sound memory results in Fig. 4b.

In addition, the dipole orientations in various liquids on the ferroelectric polymer layer were also examined by the molecular dynamic (MD) simulations (Supplementary Fig. 15) from which we computed the time-dependent electric dipole moments for 5 different liquids (water, acetonitrile, isopropanol, chlorobenzene, cyclohexane) placed on PVDF crystal substrate. As seen in Supplementary Fig. 15, the time-dependent dipole moments in the directions parallel to the PVDF substrate (μ_x, μ_y) fluctuate around zero and these dipole fluctuations in-plane direction are more pronounced as the liquid becomes polar, which indirectly represent the susceptibility of molecules in the liquid toward their dipole orientation by an external vertical field. Indeed, the dipole moments (μ_z) in the direction perpendicular to PVDF substrate (lying in the lower z position, see the layer configuration shown in Supplementary Fig. 15a) exhibit less-fluctuating, nonvanished time-average values whose magnitude is proportional to polarity of the liquid, consistent with the PFM results. The results show the polarity-dependent molecular dipole alignment by the interfaces arising from the vertical piezoelectric field in our LIFS device. We explicitly addressed these issues in the page 13 of the revised manuscript.

Figure 1. Architecture and working principle of the LIFS device. Finite element method (FEM) results of an LIFS AC device under a voltage bias between two in-plane electrodes showing both the direction and magnitude of the developed electric field without (d) and with (e) a top conductive layer on PVDF-TrFE.

Supplementary Figure 6. Polarity-dependent vertical electric field exerted on the PMMA/PVDF-TrFE bilayer based on a simple linear electric field model.

Figure 3. Vertically aligned ferroelectric polarization of the PVDF-TrFE film in the LIFS AC device. PFM images in height and piezoelectric amplitude contrast of a PVDF-TrFE layer in the LIFS AC device after DC poling between the two in-plane electrodes with (a) and without (b) a top conductive layer. Distinct piezoelectric amplitudes with opposite contrasts are apparent in the two overlapped regions between the top conductive layer and one of the two electrodes. (c) Schematic of the ferroelectric dipoles of a PVDF-TrFE layer developed in the two overlapped regions under DC poling between the two in-plane electrodes. (d) Amplitude profile along the horizontal line in (a) across the two overlapped regions. (e) PFM images in the piezoelectric amplitude contrast of PVDF-TrFE layers programmed with five liquids having different dielectric constants placed on the PVDF-TrFE film in an LIFS AC device.

Supplementary Figure 15. Molecular dynamics simulation for liquid on PVDF. (a) The snapshot of liquid (acetonitrile) layer sandwiched between PVDF and Si layer. (b)-(f) Electric dipole moment in each direction as a function of time for (b) water, (c) acetonitrile, (d) isopropanol, (e) chlorobenzene, and (f) cyclohexane. (g) The time-average of the magnitude of dipole moment in each direction for 5 liquids (water, acetonitrile, isopropanol, chlorobenzene, cyclohexane) on PVDF.

2. The author should provide the sound wave spectra to confirm whether the variation of sound pressure level came from the amplitude changes or the frequency shift of the sound wave, because the sensitivity of the microphones and human ears always depends on the frequency of the sound.

Response: The SPL results shown in the original manuscript were obtained by integrating SPL values over the frequency range which a human ear can detect, i.e., from 20 to 20 kHz, as explained in the experimental section. The SPL depends on both, the amplitude and frequency. In the experiments, we fixed an input frequency of 20 kHz and recorded a sound profile over the entire range of audible frequency. As expected, the resulting sound spectrum was obtained with a maximum amplitude at a frequency of 20 kHz, as shown in a new figure, Supplementary Fig. 26a. As the reviewer suggested, we included the sound wave spectra covering the entire frequency range in Supplementary Fig. 26b and provided an explanation on page 26 of the revised manuscript.

Supplementary Figure 26. Sound characterization of the LIFS AC device. (a) Amplitude of sound profile with time of five liquids having different dielectric constants at 20 KHz and 100 V. (b) SPL spectra over the whole range of audible frequency converted from the amplitude vs. time plots shown in (a) arising from the five liquids having different dielectric constants. (c) SPL values and maximum amplitude arising from the five liquids with different dielectric constants on an LIFS AC device. The device was operated at an AC frequency and voltage of 20 kHz and 100 V, respectively.

3. Figure 3 in supplementary materials shows the angle-dependent SPL of the LIFS device. Please give detailed explanation of this interesting phenomenon.

Response: The angle dependency of the SPL is an important property of thin piezoelectric films used in loudspeaker applications. It is known that a loudspeaker based on a thin piezoelectric film is beneficial owing to its angle-independent sound generation, whereas most film speakers are angle-dependent. The angle-independent SPL allows for uniform surrounding sound generation. As our LIFS device is also based on a thin PVDF-TrFE layer, the sensing and memorizing liquid in the SPL is angle-independent, as shown in the Supplementary Fig. 7e.

4. The author compared the SPL level of liquid with different polarity or LiCl solution with different concentration. However, as a sensor, those qualitative results are not sufficient. The author should find out the quantitative relationship between the SPL level and a certain physical parameter of the liquid, e.g. the relative permittivity, pH value or viscosity, density and so on. So we can understand the fundamental mechanism by building a quantitative physical model for the phenomenon and performance of devices. From the sensor point of view, a quantitative calibration curve is also essential for practical application. Therefore, I strongly recommend that the author could find out the fundamental and quantitative relationship between some of the physical parameters of the liquid and the SPL level of the devices.

Response: We appreciate the reviewer’s constructive comments on the quantitative calibration curve for sensor application. As explicitly described in the manuscript, the polarity (dielectric constant value) of a liquid is one of the most sensitive parameters that can be used for sensing and memorizing a liquid with our LIFS device. To obtain a calibration curve of the SPL values as a function of the polarities of solvents, we extensively examined 18 solvents having different polarities, and the polarity-dependent SPL results are shown in a new Fig. 2c and Supplementary Table 2. The results show that the SPL is almost linearly proportional to the dielectric constant of the solvents, and the sensitivity of an LIFS device is approximately 0.75 (dB/ ϵ_r), as obtained from the slope of the curve. We included these results in Fig. 2c and provided the explanation on page 10 of the revised manuscript.

Figure 2. (c) SPL values of 18 liquids examined with different dielectric constants, and the sensitivity of an LIFS AC device obtained from the linear slope of the curve. The device was operated at an AC frequency and voltage of 20 kHz and 100 V, respectively.

5. As known, the film bulk acoustic resonator (FBAR) based on the ferroelectric/piezoelectric thin films could also be used as a liquid sensor due to the shift of resonance frequency and amplitude induced by the change of mass loading on the top surface of the piezoelectric layer. Is there any possibility that the LIFS device demonstrated in this work are based on the mass-induced modulation of the resonance behavior of the PVDF-TrFE film? Because the results are so similar with the FBAR based devices. In addition, what is the resonance frequency of the PVDF-TrFE film in the LIFS device? The vibration of the PVDF-TrFE film could be largely amplified if the film was working at its resonance frequency.

Response: We appreciate the comments on the possible extension of our LIFS device to the film bulk acoustic resonator (FBAR). Although FBAR is associated with a piezoelectric resonance frequency that is sensitive to the mass loading placed on the layer, its principle is different from that of our LIFS device. As shown in a new figure, Supplementary Fig. 4, the vertical field was dependent on the conductivity and polarity of the top layer. In our LIFS device with a vertical electric field, the vertical field is obviously overlapped-area-dependent. SPL values arising from a polar liquid changed when the contact areas were altered, as shown in Supplementary Fig. 13. Interestingly, the electric field rarely changes when the heights of the liquid is varied, as shown in a new figure, Supplementary Fig. 14. We evaluated the SPL of the LIFS device with different heights of deionized water. As described in the manuscript, the water was confined in a reservoir with a constant contact area, and the water height was varied. Almost identical SPL values were obtained, regardless of the height of the water in

the range from a few micrometres to a few millimetres. The results imply that the ferroelectric polarization of our LIFS device is mainly governed by the interfacial properties of PVDF-TrFE/liquid, such as the contact area and dielectric properties of the liquid. To make our SPL results consistent, the reservoir had a constant contact area, as schematically shown in Fig. 2a. The constant contact area was also obtained in a tube type LIFS device. We have also found that the intrinsic resonance frequency of a PVDF-TrFE layer of approximately 50 MHz rarely affects the performance of our LIFS device, which works in the 20 to 20 kHz frequency range. We explicitly discussed this issue on pages 6,7,11, and 12 of the revised manuscript.

Supplementary Figure 13. SPL values arising from a polar liquid with different overlapped electrode areas on the LIFS AC device. The device was operated at an AC frequency and voltage of 20 kHz and 100 V, respectively.

Supplementary Figure 14. SPL values of the LIFS AC device as a function of the heights of a polar liquid. (a) Schematic and a photograph of the LIFS AC device with DI water, whose amount can be controlled by the height of a PDMS reservoir. SPL spectra (b) and values (c) arising from DI water with different heights on an LIFS AC device. The device was operated at an AC frequency and voltage of 20 kHz and 100 V, respectively.

6. Please check the words “on/off ratio” and “on/off difference” in the text, is there any difference? Or is it just spelling mistake?

Response: The difference between the states ‘on’ and ‘off’ was described in the ratio of the ‘on’ state value to that of ‘off’ state. We accordingly corrected the parts the reviewer mentioned.

7. What is the morphology and crystal structure of the PVDF-TrFE film? The authors should provide experimental data to support their conclusions on the composition, morphology, crystal structure and thickness of the materials.

Response: As the reviewer suggested, we extensively examined the morphologies of the PVDF-TrFE film before and after the sensing and memory operations, and the results are shown in a new figure, Supplementary Fig. 23. The characteristic needle-like crystalline domains are apparent with a length of approximately 400 nm, which is consistent with those observed in numerous previous studies on thin PVDF-TrFE films [*Appl. Phys. Lett.* **88**, 242908-242910 (2006)., *Curr. Appl. Phys.* **11**, e30–e34 (2011).]. The molecular orientation of the ferroelectric crystals of the PVDF-TrFE film was also revealed with a grazing incident X-ray scattering technique, and the results in Supplementary Fig. 24c indicate that the c-axis of the crystal, i.e., the chain axis, is dominantly aligned parallel to the surface and perpendicular to the vertical electric field. Based on the (110) reflection of the orthorhombic PVDF-TrFE crystals, we found that the crystallinity of the PVDF-TrFE film on the overlapped regions was higher than that on the gap regions, which did not have a vertical electric field responsible for enhancing the ferroelectric domain crystallization, as shown in Supplementary Fig. 24b. The crystalline orientation is also consistent with those reported previously including by us [*Appl. Phys. Lett.* **88**, 242908-242910 (2006)., *Nano Lett.* **16**, 334–340 (2015).]. We have discussed the molecular and microstructural morphology results on page 16 of the revised manuscript.

Supplementary Figure 23. Architecture and morphological analysis of the LIFS AC device with a PVDF-TrFE layer. (a) Cross-sectional scanning electron microscopy (SEM) image of an LIFS AC device. (b) SEM image of the surface morphology of the PVDF-TrFE

layer. A magnitude SEM image of the regions indicated by the dotted red square shows the characteristic needle-like crystalline domains with a length of approximately 400 nm. (c) TM-AFM image of the surface morphology of the PVDF-TrFE film.

Supplementary Figure 24. Crystalline structures of a PVDF-TrFE layer in the LIFS AC device operated with DI water. (a) Schematics of the cross-sectional view of the LIFS AC device with DI water on a PVDF-TrFE layer. (b) XRD patterns obtained from three different regions (left, centre, and right) of the PVDF-TrFE layer after 2 kV DC poling between the two in-plane electrodes. A higher crystallinity is clearly observed in both the left and right regions than in the centre regions. (c) 2D GIXD pattern of PVDF-TrFE layer on the LIFS AC device.

8. PFM or macroscale piezoelectric response testing should be done for measuring the vibration state of the PVDF-TrFE films with either vertically or laterally applied AC and DC voltage, which could provide useful information for the explanation of the working mechanism of the device.

Response: The vertical alignment of the d_{33} mode of the PVDF-TrFE film in our LIFS device was additionally confirmed with a piezoresponse microscope as the reviewer suggested, and the results are shown in a new Fig. 3. PFM images in piezoelectric amplitude contrast show that the piezoelectric amplitude significantly varied on the two overlapped regions, compared with the gap area. On the other hand, no piezoelectric response was observed on a PVDF-TrFE film when an in-plane field was applied without a top conductive layer. As expected from the FEM results, in which the fields of the two overlapped areas were in opposite directions (Fig. 1d and 1e), the piezoelectric responses in the two overlapped regions are opposite in amplitude, as evidenced in the amplitude profile shown in the schematic ferroelectric dipoles in the corresponding regions in Fig. 3c and 3d. As the vertical field developed in the overlapped area is polarity-dependent, the maximum piezoelectric amplitude is also polarity-dependent. The results shown in Fig. 3e clearly demonstrate that the higher the amplitude developed, the more polar the liquid is, which is consistent with the sound memory results in Fig. 4b.

Figure 3. Vertically aligned ferroelectric polarization of the PVDF-TrFE film in an LIFS AC device. PFM images in height and piezoelectric amplitude contrast of a PVDF-TrFE layer in an LIFS AC device after DC poling between the two in-plane electrodes with (a) and without (b) a top conductive layer. Distinct piezoelectric amplitudes with opposite contrasts are apparent in the two overlapped regions between the top conductive layer and one of the two electrodes. (c) Schematic of the ferroelectric dipoles of a PVDF-TrFE layer developed in the two overlapped regions under DC poling between the two in-plane electrodes. (d) The amplitude profile along the horizontal line in (a) across the two overlapped regions. (e) PFM images in the piezoelectric amplitude contrast of PVDF-TrFE layers programmed with five liquids having different dielectric constants placed on PVDF-TrFE film in an LIFS AC device.

Reviewer #3 (Remarks to the Author):

In their manuscript, J. S. Kim and co-authors report methods and devices for sensing liquids with polarity interactive ferroelectric sound. Based on solution-based manufacturing processes, the authors create functioning devices and perform electro-acoustic characterization measurements. They use various device architectures and implementations to show how their devices perform in basic modes of operation. This includes differentiating and memorizing liquids based on their polarity, as well as monitoring flow dynamics.

In the current version of the manuscript, it is unclear to what extent the presented results constitute a scientific discovery. The authors have to clearly state what is new and establish a better context with the existing literature. Technologically, while a need for integrated liquid sensing technology indeed exists, it is unclear to what extent the method discussed here can be applied to solve actual problems.

In order for the manuscript to be considered for publication in Nature Communications, the

authors should consider the questions and comments provided below and clearly state what is scientifically new while substantiating the technological relevance of their work:

- Fundamentals of LIFS: There are no literature references to fundamentals of liquid-interactive ferroelectric actuation and sound generation, as well as electro-acoustical liquid-solid interactions in the introductory paragraph. The authors should include suitable references here.

Response: We thank the reviewer for bringing this to our attention. As the reviewer suggested, we included Refs. 24-28 and 42-44, which are related to liquid interactive sound generation and electro-acoustical liquid–solid interactions.

- Can the measured, vertical ac voltage or the sound amplitude be modeled based on the physical properties of the materials involved, taking into consideration the actual device geometry? The manuscript does not present or refer to any theoretical predictions or model simulations of expected device performance.

Response: We appreciate the reviewer’s comment on the need of an electric field and/or sound simulation of our LIFS device. To address the issues raised by the reviewer, we first performed an electric field calculation based on the finite element method (FEM), with which we were able to confirm the vertical field arising from the top conductive layer; the results are shown in the new figures Figs. 1d and 1e. A typical in-plane and fringe field was developed between two in-plane electrodes on which PMMA and PVDF-TrFE layers were placed without a top conductive layer. When a top conductive layer was deposited, a vertically driven electric field developed on the two overlapped regions of the top conductive layer with the two in-plane electrodes. Notably, the vertical fields on the two overlapped areas were opposite in direction, which implies that the top conductive layer acted as a type of electric field mirror. Further experiments also revealed that the vertical field was conductive-dependent, as shown in a new figure, Supplementary Fig. 4. We employed a top PEDOT:PSS layer, whose conductivity was controlled by adding DMSO. The vertical field decreased with decreasing conductivity. We also observed that the vertical field was dependent on the ion conductivity as well as the polarity of the top layer, as explicitly shown in the manuscript. Note that the polarity-dependent vertical electric field with various polar liquids could not be simulated with FEM owing to the complexity of the system.

The electric field shared by the two insulating layers was calculated with a capacitance model where two capacitors containing PMMA and PVDF-TrFE were connected in series. The portion of the electric field exerted on either PVDF-TrFE or PMMA depends on both the dielectric constants and thicknesses of the layers. Based on the dielectric constants and the thicknesses of the PVDF-TrFE (5 μm) and PMMA (1.2 μm) layers, we were able to calculate the electric field exerted on the PMMA layer, which was approximately 30 % of the applied electric field. The electric field distribution of a device with a PMMA/PVDF-TrFE bilayer clearly shows the presence of a vertical electric field in the PVDF-TrFE layer, although the field strength is somewhat reduced owing to the presence of the PMMA layer, as shown in Supplementary Fig. 5. Finally, we were able to demonstrate the polarity-dependent vertical electric field exerted on both the PMMA and PVDF-TrFE layers with a simple linear electric field model as shown in Supplementary Fig. 6. We have added this explanation on pages 6 and 7 of the revised manuscript.

Figure 1. Architecture and working principle of the LIFS device. Finite element method (FEM) results of the LIFS AC device under a voltage bias between two in-plane electrodes showing both the direction and magnitude of a generated electric field without (d) and with (e) a top conductive layer on PVDF-TrFE.

Supplementary Figure 4. Dependence of the SPL and vertical AC voltage on LIFS AC devices on various properties of liquids: (a) SPL values and voltages induced by a vertical electric field arising from PEDOT:PSS with different conductivities controlled by adding DMSO on the LIFS AC device; (b) SPL values and voltages induced by a vertical electric field arising from six liquids with different dielectric constants on the LIFS AC device; and (c) SPL values and voltages induced by a vertical electric field arising from different ionic conductivities controlled by the amount of LiCl in DI water on the LIFS AC device. All the devices were operated at an AC frequency and voltage of 20 kHz and 100 V, respectively.

Supplementary Figure 6. Polarity-dependent vertical electric field exerted on a PMMA/PVDF-TrFE bilayer based on a simple linear electric field model.

- Differentiation between liquids: Could the authors explain how the measured electrical or sound signal enables a unique and unambiguous identification of the chemical identities of the tested liquids?

Response: We appreciate the reviewer's comments. As our LIFS device is based on a polarity-dependent electric field that develops upon exposing the device to AC, it is not trivial to distinguish two liquids with similar polarities, as described in the manuscript. Complementary characterization tools, including FT-IR, HPLC, and NMR, should be employed in addition to the sensing with an LIFS device to clearly identify the two liquids. To address the issues of sensitivity of LIFS devices as a function of polarities of solvents, we extensively examined a variety of solvents having different polarities (dielectric constant values), and the polarity-dependent SPL results are shown in a new figure, Fig. 2c, and Supplementary Table 2. The results show that the SPL is almost linearly proportional to the dielectric constant of the solvents and the sensitivity of an LIFS device is approximately 0.75 (dB/ ϵ_r), as obtained from the slope of the curve. Based on the master curve we obtained, we are able to predict the polarity information of an unknown liquid or a liquid with an unknown ingredient. Additional characterization should be performed for further identification. We hope that the reviewer understands that the development of a single sensing device with capability of identifying an unknown ingredient dissolved on a solvent is rarely possible. We have explained this on page 10 of the revised manuscript.

Figure 2. (c) SPL values of 18 liquids examined with different dielectric constants, and the sensitivity of the LIFS AC device obtained from the linear slope of the curve. The device was operated at an AC frequency and a voltage of 20 kHz and 100 V, respectively.

No	Solvent	Polarity index	Dielectric constant (ϵ_r)	SPL (dB)	PVDF-TrFE solvent
1	Cyclohexane	-0.2	2.02	0	
2	Mineral oil	0.0	2.30	0	
3	n-Hexane	0.1	1.88	0	
4	Trichloroethylene	1.0	3.40	0	
5	f-Propyl ether	2.4	3.90	2.03	
6	Chlorobenzene	2.7	5.60	10.32	
7	Octoxynol-9	3.2	7.50	15.31	
8	n-Octanol	3.4	10.30	20.22	
9	n-butanol	3.9	17.50	29.23	
10	Iso-propanol	4.1	20.30	32.22	
11	Ethanol	4.3	24.60	34.52	
12	Benzonitrile	4.8	25.20	35.01	
13	Methanol	5.1	32.70	38.50	
14	Acetonitrile	5.8	37.50	40.85	
15	Dimethylformamide	6.4	36.70	-	○
16	Ethylene glycol	6.9	37.70	41.24	
17	Dimethylsulfoxide	7.2	47.00	-	○
18	DI water	10.2	78.20	66.54	

Supplementary Table 2. Characteristics of 18 solvents having different polarities and dielectric constants, and their polarity-dependent SPL values on the LIFS AC device.

- Considering potential applications in microfluidics, can the authors comment on the ultimate spatial resolution that can be achieved with their method; for example, in the measurement of the 2-D position of a liquid droplet (minimum droplet volume?) on a thin pad-type device? Integration and scaling properties of the technology should be discussed.

Response: We appreciate the reviewer's insightful comments on the integration and scalability of our LIFS device. Although the focus of the present work is demonstrating the proof-of-concept of a novel sound based liquid sensing and memory device, we agree that the integration and scalability of the device should be considered for further development, as the reviewer pointed out. To address the raised issue, we systematically examined the performance of the device by controlling the size of the water droplet that can be used with the device with a fixed in-plane electrode gap of 0.2 mm. The results in the new figure Supplementary Fig. 16 show that our LIFS device successfully operated with a water droplet with of minimum diameter approximately 1 mm. Further scaling-down of our device can be achieved by reducing the gap between the two in-plane electrodes. The FEM simulation results shown in the new figure Supplementary Fig. 17 suggest that a vertical electric field developed in our LIFS device when the gap was 2 μm , below which in-plane fringe field became dominant over the vertical field. As the reviewer states in the comments that follow, the resolution of our pad-type device can be further improved with a high-performance microphone system. Note also that the physical sensing and memory area of a pad-type device may not be critical for miniaturization of the system, considering that most microfluidic systems involve rather bulky fluidic control parts, such as reservoirs, pumps, and channels. We discussed this issue on page 13 of the revised manuscript.

Supplementary Figure 16. Optimization of spatial resolution by controlling the size of the water droplet in the LIFS device. (a) Schematic of a thin pad-type LIFS device with a fixed in-plane electrode gap of 0.2 mm. (b) Photographs of the LIFS device with differently sized water droplets on the electrode gap. (c) SPL values of the LIFS AC device as a function of the droplet diameter. The device was operated at an AC frequency and a voltage of 20 kHz and 100 V, respectively.

Supplementary Figure 17. Finite element method (FEM) results of the LIFS AC device under a voltage bias between two in-plane electrodes showing the magnitude of the generated electric field with a top conductive layer on PVDF-TrFE. The vertical electric field was successfully developed when the gap was 2 μm.

- What is the benefit of using the sound from the ferroelectric vibration to identify the liquids, instead of using the vertical ac voltage? As the sound signal decreases with electrode size, this could severely limit the miniaturization and integration of sensors reported here. Are the authors considering integration of local sound sensors in order to apply and scale within a

technological context? The applicability of the method in a biomedical context seems limited due to the reliance on an external microphone.

Response: We appreciate the useful comments made by the reviewer. One of the main benefits of our LIFS device lies in the fact that sensing and memory of a liquid can be achieved with only one human sense – hearing – which makes our device human-interactive, as described in the introduction of our manuscript. The utilization of sound, rather than electric signals, can also be beneficial because sound does not require physical contact of the detection components and can readily propagate in space. This can allow for wireless detection of the liquid information and thus make it suitable potentially for on and in-body applications where a detection microphone can be remotely placed. As the reviewer is aware, the physical sensing and memory area of a pad-type device may not be critical for miniaturization of the system, as most microfluidic systems involve rather bulky fluidic control parts, such as reservoirs, pumps, and channels, as described above. It should also be noted that the demonstration of SPL sensing of a serum fluid in a capillary channel with a width and height of 7 and 3 μm , respectively, suggests the potential miniaturization of our system with an external remote microphone.

Utilization of audible sound with a frequency ranging from 20 to 20 kHz is also beneficial as our LIFS device applies to the microphone in a commercialized cell phone. Furthermore, the frequency range of audible sound we utilized for piezoelectric vibration for sensing an object is also useful for tactile surface sensing, allowing for dual-mode sensing of an object with sound and touch. An example is shown in our preliminary results below. When a human finger, which is naturally conductive, touched the overlapped area of our sensing platform with two in-plane electrodes, which is similar to LIFS with a finger instead of a liquid, both sound and tactile friction were simultaneously detected. Although the example is not directly relevant to the present LIFS work, the utilization of audible sound can be justified with our on-going efforts to develop multi-mode human interactive sensing devices. Although we did not include the preliminary results of the dual sensing platform in the Supporting Information, we have explained this issue on pages 10 and 11 of the revised manuscript.

Supplementary Figure. Multi-mode human interactive sensing device. (a) Schematics of a sensing platform with the two in-plane electrodes operated under an AC field for sound and tactile friction with a conductive human finger. (b) A schematic and a photograph of dual mode sensing with sound and touch. SPL values arising from sound when a finger touched the device as a function of the frequency at a voltage of 200 V (c) and the voltage at a frequency of 20 kHz (d). (e) Threshold voltage values arising from tactile friction when a finger was swept on the surface of the device as a function of frequency at a voltage of 200 V. The threshold voltage was determined by the minimum voltage that humans needed to could detect tactile friction. (f) Sensitivity values of tactile friction as a function of the voltage at a frequency of 100 Hz. Sensitivity was determined by the degree of tactile friction felt by a human.

- The authors state on page 11 of the main manuscript that conventional, physical pixel arrays are not required in their method. How is the sound signal from individual sensors generated, identified and differentiated in case multiple sensors are used simultaneously and need to be read out in parallel, for example, in the case of a pad-type device sensing a two-phase liquid (oil/water) where spatial differentiation between multiple liquids/droplets is required?

Response: We appreciate the reviewer's insightful comments. It is indeed not trivial to identify the locations of multiple droplets placed on our 2D pad-type LIFS device. In fact, we are investigating this issue for further technological implementation. Combined SPL values arising from the two droplets located on different positions will come out from our pad device. In principle, the positions of two droplets can be identified when all the combined SPL values are carefully assigned with sufficient margin in the SPL. At the present state of development, we are not able to realize a pad-device capable of detecting multiple positions. Alternatively, multi-droplet sensing can be achieved by physically combining more than two pad-devices, each of which can detect one type of liquid. In other words, two pad-devices for water and alcohol are independently fabricated and combined with each other. In this case, each pad-device utilizes its own frequency range, making the resulting SPL value range unique for each frequency range.

To demonstrate the proof-of-concept of our liquid detection pad, we fabricated four sets of 3×3 liquid position detection pads for single and multi-droplet position detection. The four sets of pad devices with nine position-dependent SPL values were programmed for four different liquids, i.e. DI water, ethanol, a PEDOT solution, and an LiCl solution, as shown in Fig. 5. Each 3×3 array pad was programmed with nine different DC voltages ranging from 1.0 to 1.40 kV to develop nine different marking spots with different remnant polarization values. As SPL values arising from the nine different remnant polarization values were also dependent on the AC frequency (Fig. 2e), the four different AC frequencies of 14, 16, 18, and 20 kHz were applied to zones 1, 2, 3, and 4, respectively, during the reading process, as shown in Fig. 5c. For instance, when a water droplet with a constant volume was deposited on one of the 3×3 spots of zone 2, a characteristic SPL with a maximum amplitude at 16 kHz was obtained depending on the position of the droplet, as shown in Fig. 5c. On the other hand, when an ethanol droplet with a constant volume was deposited on one of the 3×3 spots of zone 1, a characteristic SPL with a maximum amplitude at 14 kHz was obtained depending on the position of the droplet, as shown in Fig. 4c. Four sets of optically transparent, 3×3 arrays of LIFSs programmed and read by four different liquids were successfully developed as shown in Fig. 5d. The multi-droplet sensing was achieved when four droplets of four different liquids were deposited on the corresponding pads, and four different frequency readings were obtained, as shown in Fig. 4e. The four droplets of ethanol (1–2), DI water (2–4), 5 wt% LiCl (a.q.) (3–6), and the PEDOT:PSS solution (4–8) gave rise to characteristic SPL values with different maximum amplitudes at 14, 16, 18, and 20 kHz, as shown in Fig. 4e. Again, it should be noted that we are not able to detect two droplets of the same liquid on a 3×3 array device. We replaced the previous Fig. 5 with a new main figure showing these results. A detailed explanation is provided on pages 17 and 18 of the revised manuscript.

Figure 5. Position detection of a liquid droplet by 2D programmed LIFS. (a) Schematics of four sets of 3×3 liquid position detection pads for single and multi-droplet position detection. A schematic and a photograph of a single LIFS AC device pixel with PDMS space. (b) Table showing the position marking process. Each 3×3 array pad was programmed with nine different DC voltages ranging from 1.0 kV to 1.40 kV to develop nine different marking zones with different remnant polarisation values. Zones 1, 2, 3, and 4 were programmed with Ethanol, DI water, 5 wt% LiCl (a.q.) and PEDOT:PSS solution, respectively. The four different AC frequencies of 14, 16, 18 and 20 kHz were applied to zones 1, 2, 3, and 4, respectively, during the reading process. (c) SPL spectra of the nine positions of each zone. Owing to the different reading frequency values of 14, 16, 18 and 20 kHz, nine positions of each 3×3 array pad were clearly resolved in the SPL. Nine different SPL values for each liquid were obtained, depending on the position, allowing for sound-based position detection of a liquid. (d) A photograph of the position detection pad with Ethanol (1–2), DI water (2–4), 5 wt% LiCl (a.q.) (3–6) and PEDOT:PSS solution droplets (4–8). (e) SPL spectra arising from the four droplets on the position detection pad. All SPL values were obtained at a voltage of 100 V.

- Can the authors explain how the result in Supplementary Fig. 2d relates to Fig. 1 c of the main manuscript that demonstrates a voltage drop for DI water in the same frequency

range?

Response: Figure 1c clearly shows the vertical voltage drop at a frequency of approximately 10 kHz for all liquids examined with our LIFS device. The results indicate that the voltage drop at a certain frequency can be ascribed to a decrease in the dielectric constant of the PVDF-TrFE layer at the frequency and not to the intrinsic properties of the liquid, as shown in the Supplementary Fig. 2d. The dielectric constant of the PVDF-TrFE layer rapidly decreased while that of water rarely varied with the frequency. We explained this issue on page 6 of the revised manuscript.

- For improving clarity, the authors should clearly identify and label the voltages discussed in text and figures. For example, which voltage is plotted on the horizontal axis of Supplementary Fig. 6 a?

Response: We have corrected the labels and tried to improve the clarity of the manuscript.

- The authors should plot sound pressure level values versus polarity values for liquids in Fig. 3b to support the statement on page 9 of the main manuscript that those values are proportional.

Response: We appreciate the reviewer's constructive comments on a plot to generalize our LIFS device for liquid sensor application. To obtain a master curve of SPL values as a function of polarities of solvents, we extensively examined a variety of solvents having different polarities (dielectric values) and the polarity-dependent SPL results are shown in a new figure, Fig. 2c, and a new table, Supplementary Table 2. The results show that SPL is almost linearly proportional to the dielectric constant of the solvents, and the sensitivity of an LIFS device is approximately $0.75 \text{ (dB}/\epsilon_r)$, as obtained from the slope of the curve. We included these results in Figure 2, and an appropriate explanation is provided on page 10 of the revised manuscript.

Figure 2. (c) SPL values of 18 liquids examined with different dielectric constants, and the sensitivity of an LIFS AC device obtained from the linear slope of the curve. The device was operated at an AC frequency and a voltage of 20 kHz and 100 V, respectively.

- How large are performance variations from device to device, for example in the measurement shown in Fig. 1c? The authors should consider including statistical data for evaluation of technological relevance.

Response: We appreciate the reviewer's comment on the statistical analysis of the performance of our LIFS device. The results shown in the manuscript were obtained with the statistical average from five sets of LIFS devices. We carefully addressed this statistical issue by examining the results with additional sets of devices and included error bars, which clearly showed the variation in the data.

Supplementary Figure 3. Vertical AC voltage detection system between one of the bottom electrodes and various liquids deposited on the LIFS device. (a) Schematics of the device system for measuring the vertical voltage arising from a polar liquid on the LIFS AC device. (b) Time-resolved vertical voltage signals arising from various polar liquids on the LIFS AC device under an input AC voltage with a frequency of 20 kHz. (c) Vertically induced voltages arising from six liquids with different dielectric constants on an LIFS AC device. The device was operated at an AC frequency and a voltage of 20 kHz and 100 V, respectively. The values varied negligibly from one device to another, with very small error bars.

Reviewers' comments:

Reviewer #2 (Remarks to the Author):

In the revised manuscript, the authors have added sufficient experimental and theoretical simulation data for supporting the conclusion of the works. The results are convincing. The working mechanism of the LIFS devices could be well explained by the simulation and PFM characterization results. The revised paper are well organized and could provide a novel design for building liquid sensors with ferroelectric materials.

Therefore, I feel that the revised paper could be published on Nat. Comm. in the current form.

Reviewer #3 (Remarks to the Author):

Reviewer #3 would like to thank the authors for providing their response, including supporting data. The response, however, did only partially address with sufficient clarity the concerns raised by this reviewer. In order for their manuscript to be considered for publication in Nature Communications, the authors should address the following:

1. While the authors have attempted in their response to clarify some technical aspects of their work, they have failed to clearly state in their response what is the scientific discovery or breakthrough that merits publication of their work in Nature Communications. This needs to be adequately addressed in a revised version of the manuscript.

2. In their response the authors state "One of the main benefits of our LIFS device lies in the fact that sensing and memory of a liquid can be achieved with only one human sense – hearing – which makes our device human-interactive, as described in the introduction of our manuscript."

It appears that the authors are suggesting that this technology should be applied and used based on human hearing capabilities. By looking at the actual demonstrations of position detection of a liquid droplet by 2D programmed LIFS or the tube-type AC device, this reviewer is unable to identify the technological benefits that sound-based detection (or hearing) would have over electrical detection. Also, the authors do not provide convincing arguments with regards to technological concerns such as how local sound probes could be implemented, or how arbitrary liquids within the same device could be simultaneously measured (e.g. two droplets of the same liquid cannot be detected within the same array). As a result, it is still unclear to this reviewer to what extent the methods and devices discussed here can be applied to solve actual problems. Therefore, the authors should explicitly discuss in a revised version of the manuscript the limitations of their devices and review their claims of technological significance and potential impact accordingly.

3. In their response the authors state that "We appreciate the reviewer's comment on the statistical analysis of the performance of our LIFS device. The results shown in the manuscript were obtained with the statistical average from five sets of LIFS devices. We carefully addressed this statistical issue by examining the results with additional sets of devices and included error bars, which clearly showed the variation in the data."

Regarding the "statistical average from five sets of LIFS devices", the authors should provide in the revised version of the manuscript the actual numbers of devices in each set that led to the data and error bars in Fig. 3c. Typically, statistical analyses based on very small sample/device numbers (below 10) do not provide reliable information in a technological context.

4. The following sentence in the revised version of the manuscript is confusing and needs to be revised with the inclusion of appropriate physical units: "Liquid sensing based on electro acoustical

techniques is also promising, but the acoustic frequencies used the previous works are not audible from mega to giga, which region requires special detection facility.”

Pointwise responses to the comments of the reviewers are appended below. We hope that the modified version of the manuscript will be acceptable for publication in *Nature Communications*.

Reviewers' comments:

Reviewer #2 (Remarks to the Author)

Comment:

In the revised manuscript, the authors have added sufficient experimental and theoretical simulation datas for supporting the conclusion of the works. The results are convincing. The working mechansim of the LIFS devices could be well explained by the simulation and PFM characterization results. The revised paper are well orgnizaed and could provide a novel design for building liquid sensors with ferroelectric materials.

Therefore, I feel that the revised paper could be published on Nat. Comm. in the current form.

Response: We thank Reviewer #2 for previous comments that strengthened the manuscript. We are delighted that Reviewer #2 believes that the manuscript can be accepted in the current manuscript.

Reviewer #3 (Remarks to the Author)

Comment:

Reviewer #3 would like to thank the authors for providing their response, including supporting data. The response, however, did only partially address with sufficient clarity the concerns raised by this reviewer. In order for their manuscript to be considered for publication in Nature Communications, the authors should address the following:

1. While the authors have attempted in their response to clarify some technical aspects of their work, they have failed to clearly state in their response what is the scientific discovery or breakthrough that merits publication of their work in Nature Communications. This needs to be adequately addressed in a revised version of the manuscript.

Response: We thank the reviewer for the comment on the scientific discovery or breakthrough of our work. Besides the technical benefits of sensing and memorizing the information of a liquid with our LIFS, the work contains significant scientific contribution. First of all, we have quantitatively estimated the vertically driven AC field between one of two in-plane electrodes, between which an in-plane AC field had been applied, and either a polar liquid or solid surface. We should admit that the vertical electric field arising from the in-plane AC field was evident and successfully utilized for novel field-induced displays [*Nat. Commun.* **8**, 14964-14971 (2017); *Adv. Mater.* **29**, 1703552 (2017)]. To the best of our knowledge, we have employed, for the first time, the finite element analysis method (FEM) to confirm the development of the vertical electric field of our LIFS device. The results shown in Figs. 1d and 1e clearly reveal that the vertical electric field between a conductor and one of the electrodes was developed on the area overlapped with the electrode. The experimental results clearly show that the vertical field developed in the overlapped area depends on not only on the conductance but also the polarity of the top layer (Supplementary Fig. 4 and 13). We have also proposed a linear electric field model to describe the polarity-dependent vertical electric field exerted on both the PMMA and PVDF-TrFE layers, as shown in Supplementary Fig. 6. It is also one of the important discoveries that the liquid-polarity dependent d_{33} of a PVDF-TrFE layer aligned vertically and thus responsible for the ferroelectric polarization was evidenced with a piezoresponse microscope (Fig. 3). The

results shown in Fig. 3e clearly demonstrate that, the more polar a liquid, the higher the amplitude that develops, which is consistent with the sound memory results in Fig. 4b.

In addition, it is important to note that the dipole orientations in various liquids on the ferroelectric polymer layer were quantitatively examined by the molecular dynamic (MD) simulations (Supplementary Fig. 15) from which we computed the time-dependent electric dipole moments for 5 different liquids (water, acetonitrile, isopropanol, chlorobenzene, cyclohexane) placed on PVDF crystal substrate. The time-dependent dipole moments in the directions parallel to the PVDF substrate (μ_x, μ_y) fluctuate around zero, and these dipole fluctuations in-plane direction are more pronounced as the liquid becomes polar, which indirectly represent the susceptibility of molecules in the liquid toward their dipole orientation by an external vertical field. Indeed, the dipole moments (μ_z) in the direction perpendicular to PVDF substrate (lying in the lower z position, see the layer configuration shown in Supplementary Fig. 15a) exhibit less-fluctuating, nonvanished time-average values whose magnitude is proportional to polarity of the liquid, consistent with the PFM results. The results show the polarity-dependent molecular dipole alignment by the interfaces arising from the vertical piezoelectric field in our LIFS device.

Another scientific contribution of our work is that we were able to develop the quantitative relation of SPL of our LIFS as a function of the polarity (or dielectric constant) of a solvent. The 18 solvents with different polarities were employed to reveal this relationship. The results in Fig. 2c and Supplementary Table 2 show that the SPL is almost linearly proportional to the dielectric constant values of the solvents, and the sensitivity of an LIFS device is approximately 0.75 (dB/ ϵ_r), as obtained from the slope of the curve. We have briefly summarized the scientific contribution of our work in the page 23 of the revised manuscript.

2. In their response the authors state “One of the main benefits of our LIFS device lies in the fact that sensing and memory of a liquid can be achieved with only one human sense – hearing – which makes our device human-interactive, as described in the introduction of our manuscript.”

It appears that the authors are suggesting that this technology should be applied and used based on human hearing capabilities. By looking at the actual demonstrations of position detection of a liquid droplet by 2D programmed LIFS or the tube-type AC device, this reviewer is unable to identify the technological benefits that sound-based detection (or hearing) would have over electrical detection. Also, the authors do not provide convincing arguments with regards to technological concerns such as how local sound probes could be implemented, or how arbitrary liquids within the same device could be simultaneously measured (e.g. two droplets of the same liquid cannot be detected within the same array). As a result, it is still unclear to this reviewer to what extent the methods and devices discussed here can be applied to solve actual problems. Therefore, the authors should explicitly discuss in a revised version of the manuscript the limitations of their devices and review their claims of technological significance and potential impact accordingly.

Response: We appreciate the comments on the technological issues of (1) the benefit of sound-based detection, (2) how local sound probes could be implemented, (3) how arbitrary liquids within the same device could be simultaneously measured and (4) what extent the methods and devices discussed here can be applied to solve actual problems.

(1) The benefit of the sound based detection: As the reviewer is aware, there are numerous acoustic based detection technologies representatively utilizing the bulk acoustic wave (BAW)

as well as surface acoustic wave (SAW). Both types of technologies based on a piezoelectric resonance frequency sensitive to the targeting elements placed on the layer have been widely used for sensing pressure, humidity, temperature, mass as well as a variety of chemical vapors and liquids. Although the principle of devices based on BAW and SAW is different from that of our LIFS device, all of these technologies can be categorized into sound-based detection systems. As we described in the previous response to the reviewer, the utilization of sound, rather than electric signals, can also be beneficial because sound does not require physical contact of the detection components and can readily propagate in space. More beneficially, the sound from our LIFS based on thin film piezoelectricity is angle-independent (Supplementary Figure S7e). This can allow for wireless detection of the liquid information and thus make it suitable potentially for on and in-body applications where a detection microphone can be remotely placed. The discussion was made on the pages 21,22 of the revised manuscript.

(2) How local sound probes could be implemented: For wearable or on-body applications, we believe that the sound of a single liquid from an LIFS can be readily detected with the microphone in a commercial cell-phone located at a fixed position from an LIFS, as conceptually illustrated in a new Supplementary Fig. 26. For 2D pad applications, a local microphone should be implemented as the reviewer mentioned. The implementation of a microphone which may make the pad-type device bulky and complicated may not be critical for miniaturization of the system, as most microfluidic systems involve rather bulky fluidic control parts, such as reservoirs, pumps, and channels. For accurate detection, at least 4 microphones need to be placed at each edge of a pad-type device. The discussion was made on the pages 18,19 of the revised manuscript.

(3) How arbitrary liquids within the same device could be simultaneously measured: While revising the manuscript, we have found a way where we are able to detect two or more liquid droplets of a liquid in the same device by controlling the reading frequency. As shown in Fig. 5, we utilized the 4 different reading frequencies of 14, 16, 18 and 20 kHz to detect the positions of 4 different liquids of ethanol, DI water, 5 wt% LiCl and PEDOT:PSS solution accordingly placed on the zone 1, 2, 3 and 4, respectively. As described, each zone consists of 9 pixels on which the position of a liquid droplet is identified with a single AC generator. To detect multiple droplets of a liquid on a single 2D pad device, we simply designed 2 X 2 arrays of LIFSs with 4 AC generators, as schematically shown in a new Supplementary Fig. 25a. When we again employed the 4 reading frequencies of 14, 16, 18 and 20 kHz right after one step polarization poling programming instead of 9 different poling programming steps in Fig. 5, 4 different SPL in each frequency were independently obtained without interference, allowing for the detection of all the combinations of a PEDOT:PSS solution of single, two, three and four droplets on the pad device, as shown in Supplementary Fig. 25. Again, for the detection, a microphone should be implemented near the pad device. We should also admit that one pixel-one AC generator system demonstrated for multiple droplet sensing and memory can make our pad device a little bulky. We did our best to explicitly address the reviewer's concerns on technical issues with the limitations of our LIFS and its technological significance and potential impact, as the reviewer suggested. The discussion was made on the pages 18,19 of the revised manuscript.

(4) What extent the methods and devices discussed here can be applied to solve actual problems: As the Reviewer pointed out, we completely agree that the wide range of actual or potential applications for LIFSs need to be developed to drive research in this field forward. Better understanding of LIFS device structures and liquid polarity will lead to greater control over their detection capability with 'omnidirectional sound-output', and also help to increase research demands, possibly to levels that will surpass those of displays (colour-output) and

touch-based sensors (voltage-output). We think that this presents a solution for safety-critical environments, particularly from liquid detection viewpoint, where the timely detection of liquid is crucial (before it transforms to gas phase). Hazardous and toxic solvents and volatile organic compounds (VOCs) are the example of this category. In particular, VOCs have a high vapour pressure (low boiling point) and thus causes large number of molecules to evaporate from the liquid. We believe that the use of LIFSs to alert harmful liquids with omnidirectional sounds ahead of gas phase of the liquids will continue, and any increase in supply will certainly lead to new applications. The discussion was made on the page 24 of the revised manuscript.

Supplementary Figure 25. Position detection of multi-droplets of a liquid on an LIFS. (a) A schematic of 2×2 liquid position detection pad for multi-droplet position detection. (b) SPL spectra of the four positions of each zone. Owing to the different reading frequency values of 14, 16, 18 and 20 kHz, four positions of each 2×2 array pad were clearly resolved in the SPL. Four different SPL spectra for a PEDOT:PSS liquid droplet were obtained, depending on the position, allowing for sound-based position detection of a liquid. (c-e) SPL spectra arising from the two, three and four PEDOT:PSS liquid droplets on the position detection pad. All SPL values were obtained at a voltage of 100 V.

Supplementary Figure 26. A schematic of the sound detection of a liquid droplet on an LIFS with the microphone in a commercial cell-phone.

3. In their response the authors state that “We appreciate the reviewer’s comment on the statistical analysis of the performance of our LIFS device. The results shown in the manuscript were obtained with the statistical average from five sets of LIFS devices. We carefully addressed this statistical issue by examining the results with additional sets of devices and included error bars, which clearly showed the variation in the data.”

Regarding the “statistical average from five sets of LIFS devices”, the authors should provide in the revised version of the manuscript the actual numbers of devices in each set that led to the data and error bars in Fig. 3c. Typically, statistical analyses based on very small sample/device numbers (below 10) do not provide reliable information in a technological context.

Response: We are regretful for the insufficient statistical data sets shown in Supplementary Fig. 3c. We examined 5 sets of LIFS devices mainly because our results from a device to another were not significantly varied. As the reviewer suggested, we extended the device sets and examined 5 more sets of LIFS devices. The results are shown in a new Supplementary Fig. 3. The detailed quantitative numbers were also summarized in a new Supplementary Table 3, as the reviewer suggested.

Supplementary Figure 3. Vertical AC voltage detection system between one of the bottom electrodes and various liquids deposited on the LIFS device. (a) Schematics of the device system for measuring the vertical voltage arising from a polar liquid on the LIFS AC device. (b) Time-resolved vertical voltage signals arising from various polar liquids on the LIFS AC device under an input AC voltage with a frequency of 20 kHz. (c) Vertically induced voltages arising from six liquids with different dielectric constants on an LIFS AC device. The device was operated at an AC frequency and a voltage of 20 kHz and 100 V, respectively. The values varied negligibly from one device to another, with very small error bars.

Supplementary Table 3. Vertically induced voltages arising from six liquids with different dielectric constants on an LIFS AC device. The device was operated at an AC frequency and a voltage of 20 kHz and 100 V, respectively.

No	Mineral Oil	Triton-X	Ethanol	DI Water	5wt% LiCl(a.q)	PEDOT:PSS
1	1	19.4	34	40.1	45	47
2	2	21.2	34.6	40.5	44.9	46.9
3	2	19.5	34	41.1	44.8	47
4	2	22.4	33.4	40.9	45.2	47.1
5	1	19.2	33.2	39.5	45.1	47
6	1.5	20.4	32.8	40.8	43.8	46.8
7	2.5	23.5	34.2	41.2	44.2	48.1
8	2.7	18.7	33.8	40.4	45.2	47.9
9	2.2	21.9	34.4	38.9	45.5	46.5
10	1.9	17.9	32.7	40.2	44.1	45.9

4. The following sentence in the revised version of the manuscript is confusing and needs to be revised with the inclusion of appropriate physical units: “Liquid sensing based on electro acoustical techniques is also promising, but the acoustic frequencies used the previous works are not audible from mega to giga, which region requires special detection facility.”

Response: We are regretful for an ambiguous sentence. We revised the sentence accordingly with the appropriate physical units, as the reviewer suggested.

REVIEWERS' COMMENTS:

Reviewer #3 (Remarks to the Author):

I would like to thank the authors for providing their response, including additional data, which has improved clarity. I recommend acceptance of the revised version of the manuscript for publication in Nature Communications.